# Finding genetically-supported drug targets for Parkinson's disease using Mendelian randomization of the druggable genome

Catherine S. Storm [1], Demis A. Kia[1], Mona M. Almramhi[1,2], Sara Bandres-Ciga[3], Chris Finan [4,5,6], International Parkinson's Disease Genomics Consortium (IPDGC)*, Aroon D. Hingorani [4,5,7] & Nicholas W. Wood [1✉]

Parkinson's disease is a neurodegenerative movement disorder that currently has no disease-modifying treatment, partly owing to inefficiencies in drug target identification and validation. We use Mendelian randomization to investigate over 3,000 genes that encode druggable proteins and predict their efficacy as drug targets for Parkinson's disease. We use expression and protein quantitative trait loci to mimic exposure to medications, and we examine the causal effect on Parkinson's disease risk (in two large cohorts), age at onset and progression. We propose 23 drug-targeting mechanisms for Parkinson's disease, including four possible drug repurposing opportunities and two drugs which may increase Parkinson's disease risk. Of these, we put forward six drug targets with the strongest Mendelian randomization evidence. There is remarkably little overlap between our drug targets to reduce Parkinson's disease risk versus progression, suggesting different molecular mechanisms. Drugs with genetic support are considerably more likely to succeed in clinical trials, and we provide compelling genetic evidence and an analysis pipeline to prioritise Parkinson's disease drug development.

[1] Department of Clinical and Movement Neurosciences, University College London Queen Square Institute of Neurology, London, UK. [2] Department of Medical Laboratory Technology, Faculty of Applied Medical Sciences, King Abdulaziz University, Jeddah, Kingdom of Saudi Arabia. [3] Laboratory of Neurogenetics, National Institute on Aging, Bethesda, MD, USA. [4] Institute of Cardiovascular Science, Faculty of Population Health, University College London, London WC1E 6BT, UK. [5] University College London British Heart Foundation Research Accelerator Centre, New Delhi, India. [6] Department of Cardiology, Division Heart and Lungs, University Medical Center Utrecht, Heidelberglaan 100, 3584 CX Utrecht, the Netherlands. [7] Health Data Research UK, 222 Euston Road, London, UK. *A list of authors and their affiliations appears at the end of the paper. ✉email: n.wood@ucl.ac.uk

Parkinson's disease (PD) is a neurodegenerative movement disorder that currently has no disease-modifying treatment. Despite efforts, around 90% of drugs that enter clinical trials fail, mostly due to insufficient efficacy or safety[1–3]. This contributes to the staggering $1.3 billion mean price of bringing a new drug to the market[1].

Incorporating genetics in drug development could be one of the most efficient ways to improve the process, because drugs with genetic support are considerably more likely to succeed in clinical trials[4–6]. "Druggable" genes encode proteins that have been targeted by medications or are possible to target with a small molecule or monoclonal antibody[7,8]. While genome-wide association studies (GWAS) have effectively identified single nucleotide polymorphisms (SNPs) linked to PD risk and progression[9–11], the GWAS design cannot reliably pinpoint causal genes and directly inform drug development.

Mendelian randomization (MR) is a genetic technique that can predict the efficacy of a drug by mimicking a randomized controlled trial[12–15]. SNPs associated with expression levels of a gene (expression quantitative trait loci, eQTLs) may be analogous to lifelong exposure to a medication targeting the encoded protein[8,16]. The association between the same genetic variants and a disease (the outcome) can then be extracted from a GWAS for the outcome (Fig. 1a). The SNP-gene-expression and SNP-disease associations can be combined using MR to infer the causal effect of the exposure on the outcome. Since the exposure and outcome can be measured in two independent cohorts, openly available data from two large-scale GWASs can be used for one well-powered MR study. Because of Mendel's law of independent assortment, individuals are "randomized" at conception to have genetically higher or lower expression levels of the druggable gene (Fig. 1b). Individuals are generally unaware of their genotype, so the MR study is effectively blinded.

In this study, we use eQTLs in blood and brain tissue to predict the efficacy of over 3000 drug-targeting mechanisms in two independent PD case-control cohorts and examine several PD progression markers (Fig. 1c). Where possible, we repeat the analysis using SNPs associated with circulating levels of the encoded proteins. Using large-scale, openly available data and MR techniques, we propose a list of genetically-supported drug targets for PD, including repurposing opportunities of already-licensed or clinical-phase drugs.

## Results

### Mimicking medications with expression quantitative trait loci.
The druggable genome encompasses human genes that encode drug targets, including proteins targeted by approved and clinical-phase drugs, proteins similar to approved drug targets and proteins accessible to monoclonal antibodies or drug-like small molecules in vivo[7]. The most comprehensive version to date includes 4863 genes, and we sought to identify openly available eQTL data for these genes to mimic exposure to the corresponding medications[7]. Although the transcript level is biologically a step before the protein level, expression GWASs cover many genes across the genome and provide tissue specificity. Gene expression GWAS data thus provide a very good resource for high-level screens to develop drug-targeting hypotheses.

We used eQTL data from blood (31,684 mostly European-ancestry individuals)[17] and brain tissue (1387 prefrontal cortex samples of mostly European ancestry, including 679 healthy controls, 497 schizophrenia, 172 bipolar disorder, 31 autism spectrum disorder, 8 affective disorder patients)[18]. We kept eQTLs with false discovery rate (FDR) < 0.05 and located within 5 kb of the associated gene to increase the specificity of the eQTL.

As such, eQTLs were available for 2786 and 2448 druggable genes in blood and brain tissue, respectively, and these were clumped at $r^2 = 0.2$. Compared to a lower clumping threshold, this increases the number of SNPs available per gene, which in turn improves the power to detect an effect and makes it possible to test for biases in the MR estimate (as discussed later). When clumping at $r^2 = 0.2$, SNPs are not strictly independent. We therefore used MR methods that incorporate a linkage disequilibrium matrix based on the 1000 genomes EUR reference panel in the MR analysis, which accounts for correlation between SNPs[19,20]. These methods therefore take linkage disequilibrium into account.

### Discovery phase identifies 31 potential drug targets to prevent PD.
The largest GWAS available for a PD trait studied disease risk in European-ancestry individuals, which we obtained from the International PD Genomics Consortium (IPDGC)[9]. Our discovery cohort consisted of 13,708 PD patients and 95,282 controls collected for a 2014 GWAS meta-analysis[21]. The MR effect estimate for each SNP (Wald ratio) was calculated, and where >1 eQTL was available per gene after clumping, Wald ratios were meta-analysed, weighted by inverse-variance (IVW). Genetically-determined expression of 31 genes (11 in blood only; 17 in brain tissue only; three in both blood and brain tissue) was significantly associated with PD risk in the discovery cohort at FDR < 0.05. All remained significant when clumping at $r^2 = 0.001$ (Supplementary Data 1).

### 15 potential preventative agents replicate in an independent PD case-control cohort.
We sought to replicate all genes that reached significance in the discovery phase using the Wald ratio or IVW method in an independent PD case-control cohort (Fig. 1). The replication population consisted of 8036 PD patients and 5803 controls (no overlap with the discovery cohort)[9]. The MR methods were identical to those used in the discovery phase.

Genetically-predicted expression of 15 genes (four in blood only; nine in brain tissue only; two in both tissues) replicated using the Wald ratio or IVW method (Fig. 2 and Supplementary Data 1). *BST1*, *CD38*, *CHRNB1*, *CTSB*, *GPNMB*, *LGALS3*, *MAPT*, *MMRN1*, *NDUFAF2*, *PIGF*, *VKORC1* and *WNT3* reached FDR < 0.05; *ACVR2A*, *HSD3B7* and *MAP3K12* reached nominal significance. *GPNMB* and *HSD3B7* reached significance in both blood and brain tissue. Of these 15 potential drug targets to prevent PD, nine were not nominated by the PD risk GWAS meta-analysis[9].

Three replicated genes encode targets of approved or clinical-phase drugs with an appropriate direction of effect for PD protection, presenting a possible repurposing opportunity: *CHRNB1*, *NDUFAF2* and *VKORC1* (Table 1 and Supplementary Data 1). The GPNMB protein is a receptor targeted by glematumumab, an antibody-drug conjugate that is being evaluated for several types of cancer[22]. After binding to GPNMB, the drug is internalised by the cell and is cytotoxic. Since this mechanism of action does not reflect a change in GPNMB levels, we do not consider glematumumab a potential candidate for repurposing. We find that CD38-inhibitors such as daratumumab, licensed to treat multiple myeloma, and MAP3K12-inhibitors such as CEP-1347 may increase PD risk. Interestingly, CEP-1347 failed to modify PD progression in a phase 3 clinical trial[23], and our data may provide a genetic explanation why CEP-1347 was unsuccessful.

### MR quality control suggests that *CD38*, *CTSB*, *GPNMB* and *MAP3K12* have the most robust MR evidence for PD risk.
We completed a series of quality control steps to prioritise

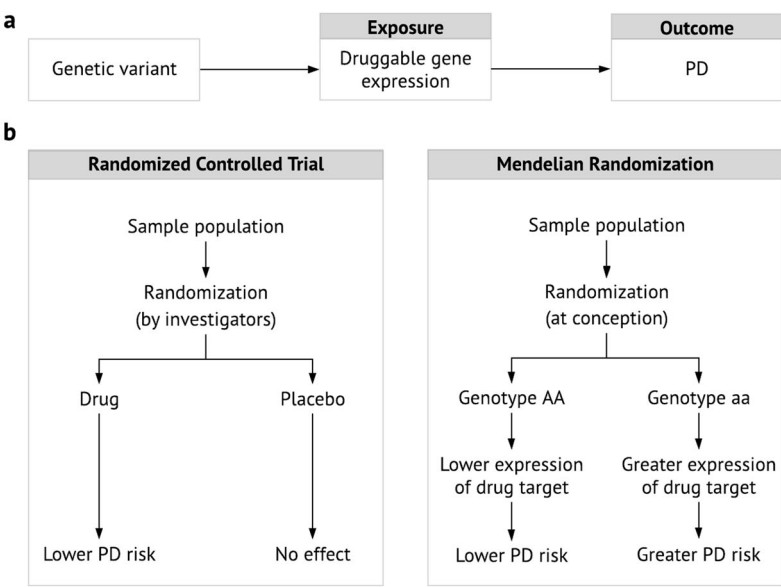

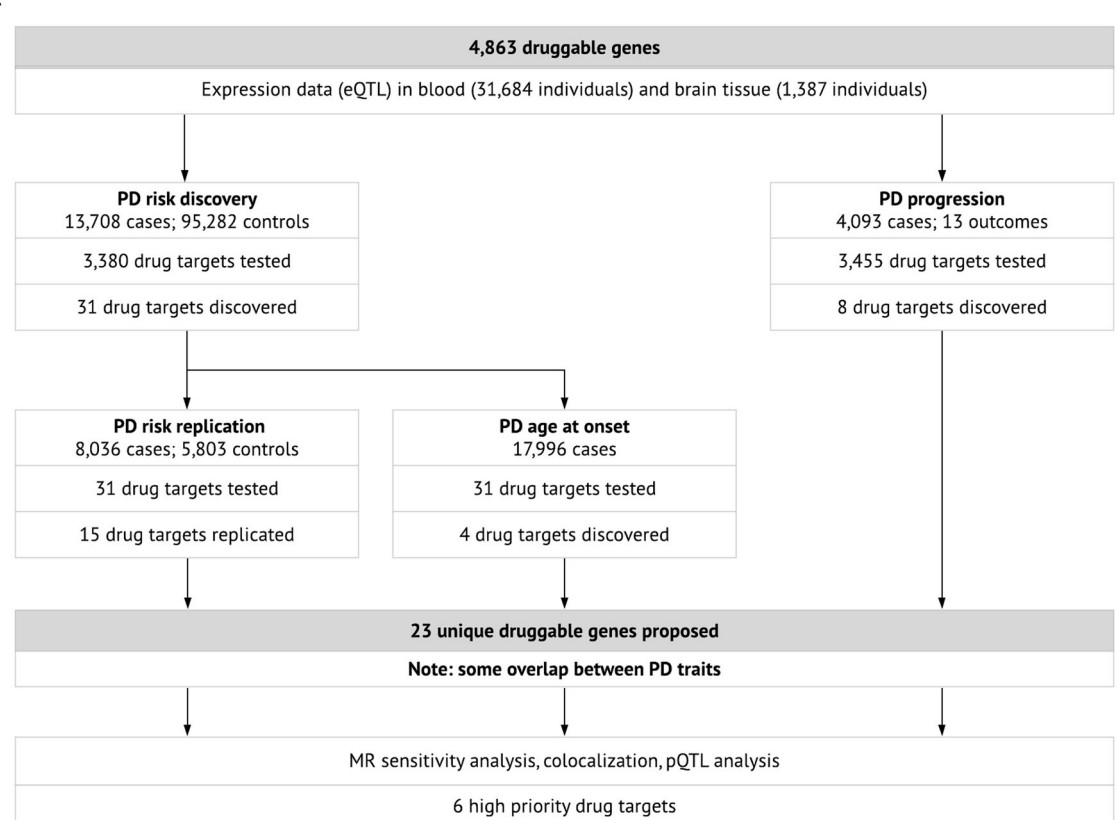

**Fig. 1 Overview of MR and our study. a** Genetic variants associated with the expression of a gene are called eQTLs, and they mimic life-long exposure to higher or lower levels of gene expression (the exposure). These variants affect PD (the outcome) through the exposure only, i.e. there is no horizontal pleiotropy. **b** MR is analogous to a randomized controlled trial, where individuals are randomly allocated to a genotype according to Mendel's law of independent assortment[14]. **c** Workflow and summarized results of our MR study. eQTL expression quantitate trait locus, MR Mendelian randomization, PD Parkinson's disease, pQTL protein quantitative trait locus.

the replicated genes. The direction of effect was consistent between the discovery and replication phases for all 15 replicated genes (Supplementary Data 2). Previous eQTL-based MR studies have reported heterogeneity in magnitude and direction of effect between tissues[8,24], and we found that raised *HSD3B7* expression was associated with raised PD risk in blood and reduced PD risk in brain tissue (Fig. 2). This pattern was consistent between the discovery and replication phase. Although this may suggest opposing effects between tissues, there was only one eQTL available for *HSD3B7* in brain and two eQTLs in blood (discovery phase). Results based on one or two SNPs should be interpreted with caution, because it is not possible to perform the additional quality control discussed below.

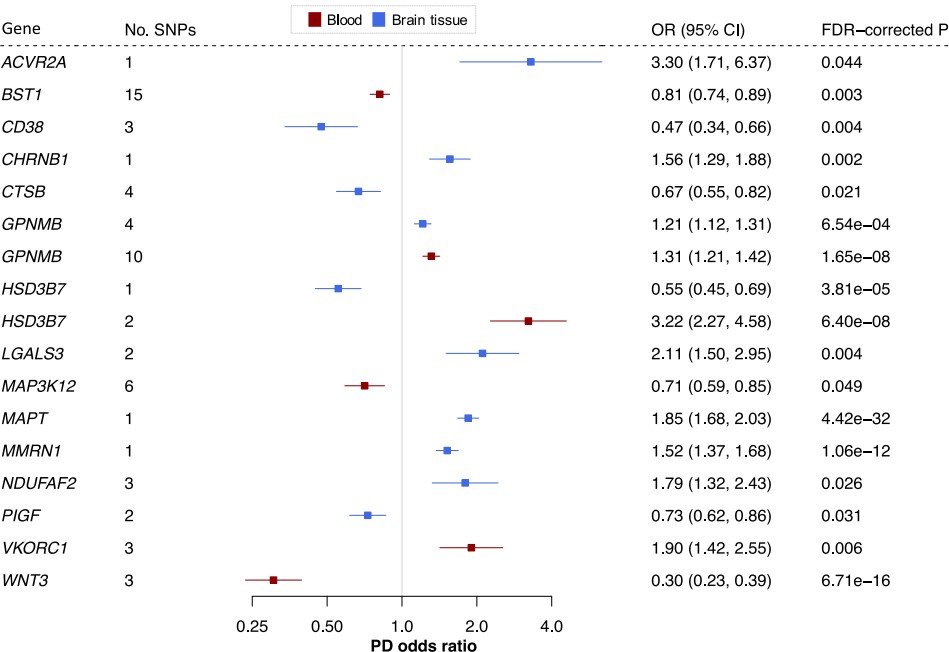

**Fig. 2 Fifteen potential preventative drug targets reach significance in two independent PD case-control cohorts.** Forest plots showing the discovery-phase results for the 15 replicated genes. The centre of the error bars represents the PD odds ratio per 1-standard-deviation increase in gene expression, calculated using the Wald ratio (if 1 SNP) or IVW (if >1 SNP) and corrected for the number of genes tested. Results are colour-coded according to the tissue (red = blood, blue = brain tissue). 95% CI 95% confidence interval, FDR false discovery rate, OR odds ratio, PD Parkinson's disease.

**Table 1 Four potential drug-targeting mechanisms for PD may constitute repurposing opportunities for existing drugs.**

| Gene | Outcome (tissue) | Drug name | Clinical phase | Indications/Uses |
|---|---|---|---|---|
| CHRNB1 | Risk (brain) | Rocuronium | Approved | Muscle relaxant in anaesthesia |
| NDUFAF2 | Risk (brain) | Metformin | Approved | Type 2 diabetes mellitus, polycystic ovarian syndrome |
| RHD | Dyskinesia (brain) | Roledumab | Phase 2 | Prevent alloimmunisation in Rhesus negative mothers carrying a Rhesus positive child |
| VKORC1 | Risk (blood) | Warfarin | Approved | Prophylactic anticoagulation *high risk of falls in Parkinson's disease |

These drugs are either approved or in clinical trial phase, and the mechanism of action is consistent with the direction of our MR effect estimate. The second column displays the potential effect on PD and target tissue. Clinical phase and drug indication based on https://clinicaltrials.gov/ and the British National Formulary. Direction of effect was confirmed using https://www.drugbank.ca or https://www.ebi.ac.uk/chembl/ databases.

The IVW method assumes that (1) the genetic variant(s) must be associated with the exposure, (2) the genetic variant(s) must not be associated with any confounders, and (3) the genetic variant(s) must not be associated directly with the outcome. This means that the SNP should affect the outcome (PD risk) through the exposure (gene expression) only, so the y-intercept of the IVW regression is fixed at zero[25]. This assumption is violated if there is genetic pleiotropy, where a SNP affects the outcome through an alternative pathway. This kind of pleiotropy may arise due to measured and unmeasured confounders, for example if the SNP is an eQTL for another gene that is not tested in this MR study. If pleiotropy pushes the effect in one direction, the IVW method yield a biased effect estimate. The MR-Egger method relaxes this assumption by not constraining the y-intercept. If the MR-Egger y-intercept significantly deviates from zero, this suggests that there is directional pleiotropy. This method assumes that any pleiotropic effects are independent of the gene-exposure association[26].

If several meta-analysis methods yield a similar result, such as the MR-Egger and maximum likelihood methods, we consider the MR result more robust[25,27,28]. The latter allows more uncertainty in the SNP-exposure and SNP-outcome associations[29]. These methods are only possible if >2 SNPs are available per gene, and all genes with >2 SNPs reached at least nominal significance using the maximum likelihood method (uncorrected $p < 0.05$). The magnitude and direction of effect were largely consistent between methods, except for BST1. For BST1, the MR-Egger estimate was in the opposite direction to the IVW and maximum likelihood results (Supplementary Data 1). All genes with >2 SNPs available passed the MR-Egger intercept test except BST1, explaining the deviant MR-Egger estimate for this gene (Supplementary Data 2).

Nevertheless, if SNPs for a gene are pleiotropic in opposing directions, the MR-Egger y-intercept will still be zero. The Cochran's Q and $I^2$ tests usefully assess overall heterogeneity between Wald ratios. NDUFAF2, WNT3 and VKORC1 did not pass the Cochran's Q ($p < 0.05$) nor $I^2$ ($I^2 > 0.50$) tests (Supplementary Data 2). This means that there is significant heterogeneity in the MR result for these genes, and such heterogeneity among Wald ratios can for example happen if at least one SNP for the gene is pleiotropic[30].

We repeated the analysis in the discovery outcome data using only SNPs that were specifically associated with our replicated genes. In other words, we removed any SNPs associated with the expression of any other gene in the original eQTL dataset. All replicated genes remained significant in this analysis (Supplementary Data 8 and 9).

In addition, a spurious MR result may arise from a locus where the SNP-exposure and SNP-outcome associations are rooted in two distinct causal SNPs in close linkage disequilibrium[30]. When the SNP is significantly associated with both exposure and outcome, this can be probed using colocalization analysis[31]. There is evidence that proteins with both MR and colocalization evidence are more likely to be successful drug targets[32]; this may simply reinforce that GWAS-nominated drug targets are more likely to reach approval[4]. Using the discovery outcome data, we had sufficient power ($PPH3 + PPH4 \geq 0.8$) to perform a colocalization analysis for 13 genes (see 'Methods' and Supplementary Data 7). Of these, *ACVR2A, BST1, CHRNB1, CTSB, GPNMB, HSD3B7, LGALS3, MAPT, MMRN1* and *VKORC1* had strong evidence of colocalization ($PPH4 \geq 0.75$). All genes with sufficient power colocalized in the replication data (*BST1, CD38, GPNMB, HSD3B7, MAPT, MMRN1, VKORC1* and *WNT3*). Similarly, Kia et al. recently found that eQTLs in brain tissue for *CD38* and *GPNMB* based on a different eQTL dataset colocalize with PD risk loci[33], strengthening the evidence for the encoded proteins as drug targets for PD.

**Four potential targets for preventative drugs may also affect PD age at onset.** Pharmacologically delaying the age of onset of a debilitating disease may have a considerable impact on both socioeconomic burden of disease and quality of life by providing disability-free years to people at risk. Evidence from polygenic risk score analyses suggests that genetic risk of PD is correlated with PD age at onset[11,34–36]. We therefore asked whether expression of the genes reaching significance in our MR discovery phase for PD risk also predict PD age of onset. We sourced openly available summary statistics from a PD age of onset GWAS, including 17,996 patients (Fig. 1c). Based on the same analysis pipeline as the replication step for PD risk, expression of four genes predicted PD age of onset at $p < 0.05$: *BST1* in blood, *CD38* in brain tissue, *CTSB* in brain tissue and *MMRN1* in brain tissue (Fig. 3 and Supplementary Data 3). *CD38* and *MMRN1* remained significant when clumping at $r^2 = 0.001$. There were >2 SNPs available for *BST1, CD38* and *CTSB*, and the IVW, maximum likelihood and MR-Egger methods yielded a consistent direction of effect for these genes (Supplementary Data 1). All three genes passed the MR-Egger intercept ($p > 0.05$), Cochran's Q ($p > 0.05$), and $I^2$ tests (($I^2 < 0.50$). *BST1* and *MMRN1* remained significant when removing SNPs associated with expression of any other gene in the original eQTL dataset

(Supplementary Data 8 and 9). Of the four genes, we had sufficient power ($PPH3 + PPH4 \geq 0.8$) to perform a colocalization analysis for *BST1*, and we found strong evidence of colocalization ($PPH4 \geq 0.75$; Supplementary Data 7).

If increased expression of a gene predicts reduced PD risk, this gene should be associated with a delayed age at onset. This was consistently the case for all four genes that reached significance for age at onset. Overall, these data suggest that there may be some shared molecular mechanisms driving PD risk and age at onset, yet this overlap may be incomplete.

**There is little overlap between drug targets to prevent PD and reduce PD progression.** The PD risk GWAS data afford large discovery and replication cohorts, which is a great advantage in MR. Nevertheless, it is currently not possible to reliably predict PD, limiting the immediate usefulness of a drug to prevent or delay this condition. Many clinical trials for PD use progression markers such as the Unified PD Rating Scale (UPDRS) to evaluate drug efficacy, and it remains unclear how the molecular mechanisms driving PD risk relate to clinical progression. We used MR to probe whether expression of any of the 4863 drug-gable genes is significantly associated with PD progression, measured by the UPDRS (total and parts 1 to 4), mini-mental state examination (MMSE), Montreal cognitive assessment (MOCA), modified Schwab and England activities of daily living scale (SEADL), Hoehn and Yahr stage, dementia, depression, and dyskinesia. The MR pipeline for each progression marker was identical to the discovery phase for PD risk (Fig. 1).

We used openly available summary statistics from a GWAS for these PD progression markers in 4093 European PD patients, followed over a median of 2.97 years[10]. 3455 druggable genes had an eQTL available for MR using a PD progression marker (2752 in blood, 2353 in brain tissue), and eight genes reached significance across five progression outcomes (Fig. 4 and Supplementary Data 1). One of these, *RHD*, encodes the target of a clinical-phase medication with an appropriate direction of effect, possibly representing a repurposing opportunity (Table 1).

*IRAK3* expression in blood was significantly associated with UPDRS parts 2 and 4, depression, and dyskinesias. *LMAN1* expression in blood reached significance for dyskinesias and UPDRS part 2. Reaching significance for several progression markers strengthens the evidence for these two genes. No genes reached significance for both PD risk and progression. Since age at motor symptom onset may be considered an early marker of

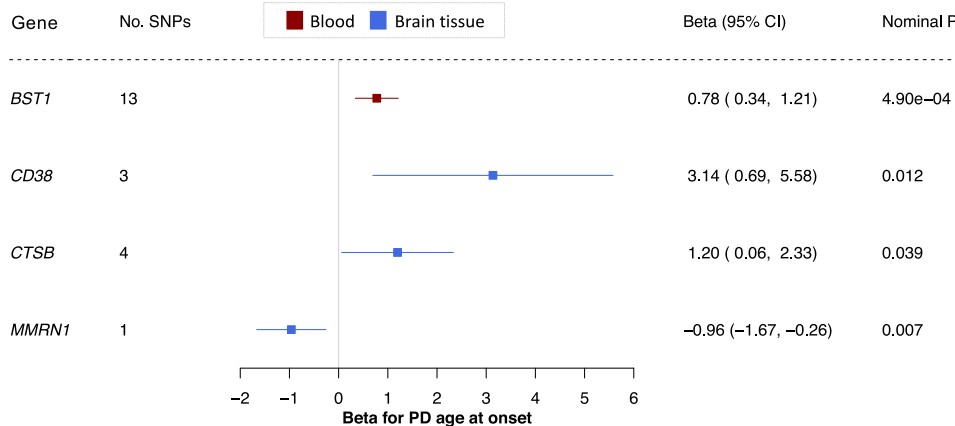

**Fig. 3 Four potential preventative drugs may also affect PD age at onset.** Forest plot; the centre of the error bars represents the standard-deviation change in PD age at onset per 1-standard-deviation increase in gene expression, calculated using the Wald ratio (if 1 SNP) or IVW (if >1 SNP) and colour-coded by tissue (red = blood, blue = brain tissue). A negative beta corresponds to a younger age at onset, and a positive beta corresponds to an older age at onset. 95% CI 95% confidence interval, PD Parkinson's disease.

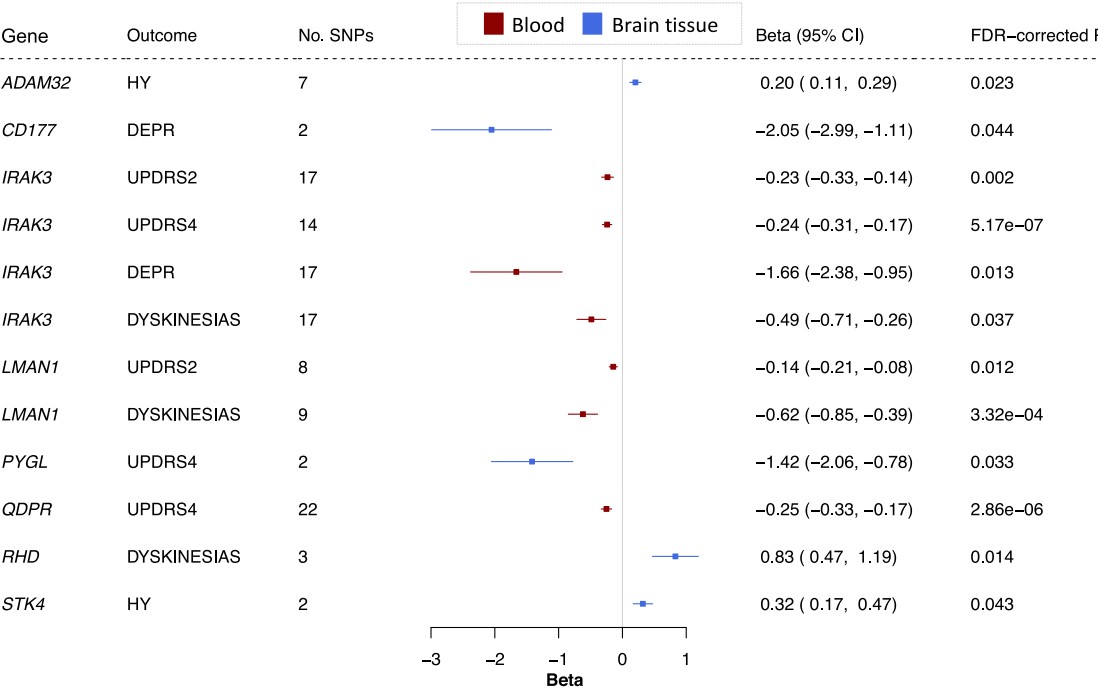

**Fig. 4 Genetically-predicted expression of eight genes in blood or brain tissue is associated with PD progression markers.** Forest plot; the centre of the error bars show the standard-deviation change in each progression marker, per 1-standard-deviation increase in gene expression, calculated using the Wald ratio (if 1 SNP) or IVW (if >1 SNP). Results are colour-coded by tissue (red = blood, blue = brain tissue) and corrected for the number of genes tested. 95% CI 95% confidence interval, DEPR depression, FDR false discovery rate, HY Hoehn and Yahr, DEPR depression, UPDRS2-4 Unified Parkinson's Disease Rating Scale parts 2 to 4.

PD progression, we also used our MR approach to assess whether genes discovered by our progression analysis causally predict age at onset. Of the genes that reached significance for a progression marker, none reached nominal significance using the age at onset data (Supplementary Data 3 and 4). *CD177, IRAK3, RHD* and *STK4* remained significant when removing SNPs associated with expression of any other gene in the original eQTL dataset (Supplementary Data 8 and 9). It was not possible to perform a reliable colocalization analysis in our progression study, since the discovered genes did not have sufficient power to do so (i.e. $PPH3 + PPH4 < 0.8$).

The direction of effect was consistent between the IVW, maximum likelihood and MR-Egger methods for all genes except *RHD*, where the MR-Egger method opposed the direction of the IVW and maximum likelihood methods. *CD177* (depression), *RHD* (dyskinesia), *PYGL* (UPDRS part 4) and *STK4* (Hoehn and Yahr) reached significance when clumping at $r^2 = 0.001$. *ADAM32* (Hoehn and Yahr), *IRAK3* (dyskinesia), *LMAN1* (UPDRS part 2), and *RHD* (dyskinesia) passed MR-Egger intercept, Cochran's $Q$ and $I^2$ tests (Supplementary Data 2). Taken together, these five genes have the most robust MR evidence for modifying a PD progression marker.

**Protein quantitative trait locus data provide further genetic evidence.** Most clinically-used drugs target proteins, not gene expression, and genetic variants associated with protein levels (protein quantitative trait loci, pQTLs) may model drug target effects more accurately than eQTLs[8]. Even with high throughput protein assays, however, the spectrum of reliable, well-powered GWAS data on protein targets is limited. Many genetic studies on protein levels are based on plasma and lack tissue diversity[37–39]. Of the 23 proposed targets, we found pQTLs for BST1, CD38, CTSB, GPNMB and LGALS3 for PD risk, as well as PYGL and QDPR for UPDRS part 4[37–40].

Our MR analysis found that BST1, CTSB and LGALS3 levels were consistently associated with PD risk ($p < 0.05$; Fig. 5 and Supplementary Data 5). The result for GPNMB (risk) and PYGL (UPDRS part 4) lost significance when using data from different pQTL studies. The direction of effect was consistent between the pQTL and eQTL results for all genes except BST1, and the MR-Egger intercept, Cochran's $Q$ and $I^2$ tests suggest that the BST1 results may be biased by genetic pleiotropy (Supplementary Data 6). This illustrates the importance of MR quality control—maximizing the number of SNPs available per drug target and validation with different data types and independent replication cohorts is essential for a reliable effect estimate.

**Discussion**
This work explicitly seeks to identify new drug targets for PD, and we provide genetic evidence in favour of 23 potential disease-modifying drug targets. Tables 2 and 3 summarize the evidence supporting these genes. The genes were prioritised using several meta-analysis methods (IVW, MR-Egger and maximum likelihood), the MR-Egger intercept test, Cochran's $Q$ test, the $I^2$ test, a pQTL study, colocalization analysis and previously published MR and colocalization evidence. This allowed us to look for pleiotropy due to both measured and unmeasured confounders[25]. We propose six drug targets with the strongest MR evidence: *CTSB, GPNMB, CD38, RHD, IRAK3* and *LMAN1*.

We identifed four genes encoding targets for existing drugs warranting further discussion (Table 1). *NDUFAF2* encodes a subunit of a target of metformin, an approved medication for type 2 diabetes mellitus. There is extensive evidence for a relationship between diabetes mellitus and PD[41], and several rodent studies have investigated the potential of metformin as a neuroprotective agent[41–43]. We found significant heterogeneity in the MR result for this gene. Although this may be because at least one SNP for this gene is pleiotropic, we speculate that this could occur if the

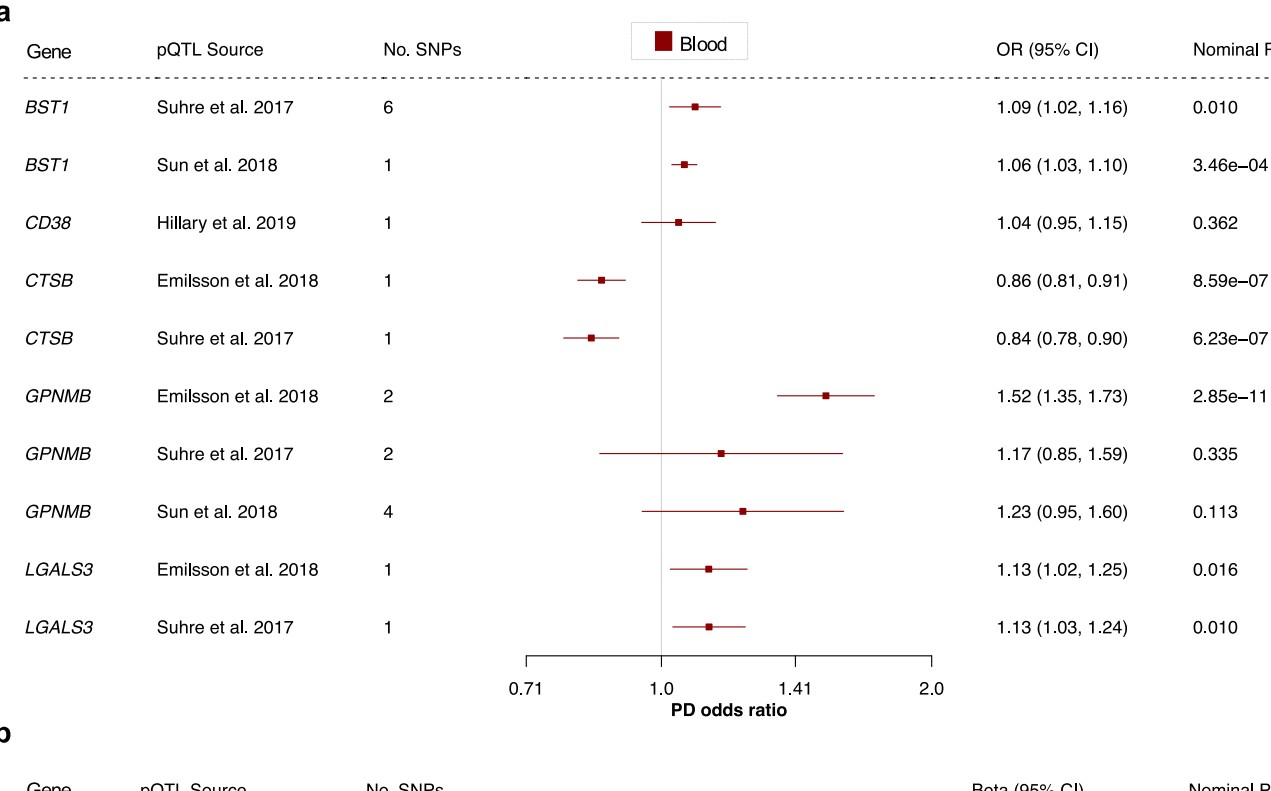

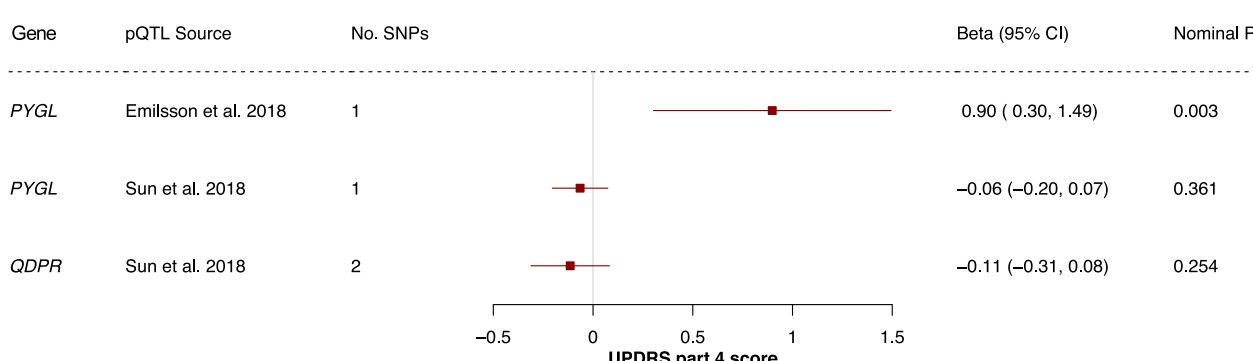

**Fig. 5 Protein quantitative trait loci in blood provide further genetic evidence.** Forest plots showing the results for all proteins and outcomes where a pQTL was available. The centre of the error bars show the (**a**) PD odds ratio and (**b**) standard-deviation change in UPDRS part 4 score, per 1-standard-deviation increase in circulating protein levels, calculated using the Wald ratio (if 1 SNP) or IVW (if >1 SNP). The "pQTL Source" column indicates which pQTL study the SNPs were derived from. 95% CI 95% confidence interval, OR odds ratio, PD Parkinson's disease, pQTL protein quantitative trait locus, UPDRS Unified Parkinson's Disease Rating Scale.

effect is driven by a subset of PD patients. This, however, remains a subject for future research, because the GWAS data used in this study are not stratified by any kind of PD subtype. Epidemiological studies on the relationship between long-term medication use and incidence of a disease are an invaluable contribution to evaluating preventative agents for PD. A retrospective cohort study of over 6000 patients with type 2 diabetes mellitus found that more than four years of metformin use maybe associated with a reduced PD incidence[44]. Our MR study thus provides further evidence in favour of repurposing anti-diabetic drugs for PD.

Other medications may not be as suitable for repurposing. To our knowledge, there is no evidence linking PD and the drug roledumab, which is currently in a phase II clinical trial to prevent alloimmunisation in Rhesus negative mothers carrying a Rhesus positive child (NCT02287896). Our evidence suggests that *RHD* expression in brain tissue, rather than blood, is associated with PD dyskinesia. Next, *CHRNB1* encodes the beta subunit of the muscle acetylcholine receptor at the neuromuscular junction,

which is inhibited by muscle relaxants used during surgical anaesthesia. *VKORC1* encodes the catalytic subunit of the vitamin K epoxide reductase, and this enzyme is targeted by the oral anticoagulant warfarin. The key adverse effect of warfarin treatment is haemorrhage, and since PD is a movement disorder where patients experience frequent falls, any potential benefit of warfarin treatment would likely be outweighed by the added risk of haemorrhagic strokes and complications of bleeding.

The two-sample MR design allowed us to explore different tissues and PD traits, and we identified different candidates to prevent, delay onset, and slow progression of PD (Figs. 2, 3 and 4). Although we found that four of the drug targets for PD risk may also affect PD age at onset, we found very different candidates for progression. Age at motor symptom onset can be considered an early sign of PD progression, and it is striking that none of the genes that reached significance for a progression outcome reached significance in the age at onset data. These results are in line with the GWAS data, finding little overlap between loci associated with PD risk, age at onset and progression

**Table 2 Evidence supporting druggable genes whose expression was significantly associated with PD risk or age at onset using the Wald ratio or IVW method.**

| Gene | PD outcome | Tissue | Replication | Sign. with max. lik. | Sign. with MR-Egger | MR-Egger intercept test | Cochran's Q test | $I^2$ test | pQTL evidence | Coloc | Previously published MR or coloc support |
|---|---|---|---|---|---|---|---|---|---|---|---|
| GPNMB | Risk | Blood | ✓ | ✓ | ✓ | ✓ | ✓ | ✓ | ✓ | ✓ | MR[9] |
|  | Risk | Brain | ✓ | ✓ | ✓ | ✓ | ✓ | ✓ |  | ✓ | MR[24]; coloc[33] |
| CTSB | Risk | Brain | ✓ | ✓ | x | ✓ | ✓ | ✓ | ✓ | ✓ | MR[9] |
|  | Age at onset | Brain |  | ✓ | x | ✓ | ✓ | ✓ |  |  |  |
| VKORC1 | Risk | Blood | ✓ | ✓ | x | ✓ | ✓ | ✓ |  | ✓ |  |
| CD38 | Risk | Brain | ✓ | ✓ | x | ✓ | ✓ | ✓ | x | x | Coloc[33] |
|  | Age at onset | Brain |  | ✓ | x | ✓ | ✓ | ✓ |  |  |  |
| MAP3K12 | Risk | Blood | ✓ | ✓ | x | ✓ | ✓ | ✓ |  | x |  |
| NDUFAF2 | Risk | Brain | ✓ | ✓ | x | ✓ | ✓ | ✓ |  | x |  |
| BST1 | Risk | Blood | ✓ | ✓ | x | x | ✓ | ✓ | ✓ | ✓ | MR[9] |
|  | Age at onset | Blood |  | ✓ | x | ✓ | ✓ | ✓ |  | ✓ |  |
| HSD3B7 | Risk | Blood | ✓ |  |  | ✓ | ✓ | ✓ |  | x |  |
|  | Risk | Brain | ✓ |  |  |  |  |  |  | ✓ | MR[24] |
| LGALS3 | Risk | Brain | ✓ |  |  |  |  |  | ✓ | ✓ |  |
| ACVR2A | Risk | Brain | ✓ |  |  |  |  |  |  | ✓ |  |
| CHRNB1 | Risk | Brain | ✓ |  |  |  |  |  |  | ✓ | MR[9] |
| MAPT | Risk | Brain | ✓ |  |  |  |  |  |  | ✓ |  |
| MMRN1 | Risk | Brain | ✓ |  |  |  |  |  |  | ✓ |  |
|  | Age at onset | Brain |  |  |  |  |  |  |  |  |  |
| PIGF | Risk | Brain | ✓ |  |  |  |  |  |  |  |  |
| WNT3 | Risk | Blood | ✓ | ✓ | x | ✓ | x | x |  | x | MR[9] |

✓ pass, x fail, *blank* not possible to test, *coloc* colocalization, *max. lik.* maximum likelihood, *MR* Mendelian randomization, *PD* Parkinson's disease, *pQTL* protein quantitative trait locus, *sign* significant.

**Table 3 Evidence supporting druggable genes whose expression was significantly associated with a PD progression trait using the Wald ratio or IVW method.**

| Gene | PD outcome | Tissue | Sign. with max. lik. | Sign. with MR-Egger | MR-Egger intercept test | Cochran's Q test | $I^2$ test | pQTL evidence | Coloc |
|---|---|---|---|---|---|---|---|---|---|
| RHD | Dyskinesia | Brain | ✓ | x | ✓ | ✓ | ✓ |  |  |
| IRAK3 | Dyskinesia | Blood | ✓ | ✓ | ✓ | ✓ | ✓ |  |  |
|  | Depression | Blood | ✓ | ✓ | x | ✓ | ✓ |  |  |
|  | UPDRS part 2 | Blood | ✓ | x | x | ✓ | ✓ |  |  |
|  | UPDRS part 4 | Blood | ✓ | ✓ | x | ✓ | ✓ |  |  |
| ADAM32 | Hoehn and Yahr | Brain | ✓ | x | ✓ | ✓ | ✓ |  |  |
| LMAN1 | UPDRS part 2 | Blood | ✓ | x | ✓ | ✓ | ✓ |  |  |
|  | Dyskinesia | Blood | ✓ | ✓ | x | ✓ | ✓ |  |  |
| PYGL | UPDRS part 4 | Brain |  |  |  |  |  |  | ✓ |
| CD177 | Depression | Brain |  |  |  |  |  |  |  |
| QDPR | UPDRS part 4 | Blood | ✓ | ✓ | ✓ | x | x | x |  |
| STK4 | Hoehn and Yahr | Brain |  |  | x | x | x |  |  |

✓ pass, x fail, *blank* not possible to test, *coloc* colocalization, *max. lik.* maximum likelihood, *MR* Mendelian randomization, *PD* Parkinson's disease, *pQTL* protein quantitative trait locus, *sign* significant, *UPDRS* Unified Parkinson's Disease Rating Scale.

markers[9–11]. This may reflect the limited sample size of current PD progression GWAS data. Nevertheless, this raises questions about what drives PD susceptibility versus progression, painting a yet unclear picture of partially overlapping molecular mechanisms.

Our candidates to slow PD progression may be of most immediate relevance, because currently PD cannot be accurately predicted. A preventative agent would need to be highly tolerable and have a very safe side effect profile, and our approach is not well suited for systematically evaluating the safety aspects of our proposed candidates in this study. To our knowledge, the data used here are from the largest openly available progression GWAS to date. We did not find any non-overlapping PD progression GWAS with sufficient power for a replication step in our progression analysis, which would have to measure progression in a similar way to the study used here. As such, the preventative list carries more robust evidence, because each gene reached significance in two large, independent cohorts. Replication is critical to validating scientific findings and eliminating false positives, and this has been an crucial lesson for genetic research[45–47]. Replication is not common practice in MR yet[48], and it is a key strength of our study. Although including all samples available in

one analysis would maximise statistical power[46,49], using independent discovery and replication cohorts allowed us to validate our proposed drug targets. Since our overarching intention was to provide genetic evidence to improve success rates in clinical trials, we made this decision to reduce the number of false positives.

Our study has valuable advantages compared to previous MR projects studying PD using QTL data. In the latest GWAS meta-analysis for PD risk in Europeans, Nalls et al. selected SNPs associated with PD risk and used MR to identify whether any of these loci alter expression or methylation of genes within 1 Mb of the SNP[9]. This contrasts with our exposure-centred MR analysis, where we chose SNPs associated with the expression of a druggable gene, rather than the disease outcome. More recently, Baird and colleagues conducted a transcriptome-wide MR study for a series of brain diseases and found six genes whose expression in brain tissue was significantly associated with PD risk[24]. Two of these were also discovered in our study: *GPNMB* and *HSD3B7*. The remaining four were either not part of the druggable genome, rendering the encoded proteins less actionable drug targets, or did not reach significance in our discovery or replication cohorts, illustrating the importance of replication. Furthermore, our MR study is the first to study druggable genes in the context of PD age at onset and progression.

Nevertheless, progression and age at onset studies are particularly affected by collider bias[50–52]. For example, if expression of a gene and depression are both associated with disease risk, that gene's expression will be artificially associated with depression in a cohort containing only cases. In a progression study, genetic variants that cause disease will thus be associated with other risk factors for disease. The druggable genes we identified in our progression study did not reach significance in our risk study, so this kind of collider bias is less likely to have occurred for our candidate genes. The age at onset analysis was comparably more affected, since we tested genes that reached significance for PD risk. Overall, this emphasises the importance of MR quality control methods (including replication) for identifying reliable causal effects, representative sampling in GWAS, as well as continued development of methods to formally test for collider bias[53,54].

Another key limitation of this study is that MR cannot fully recapitulate a clinical trial. MR mimics lifelong, low-dose exposure to a drug and assumes a linear relationship between exposure and outcome. This differs from a clinical trial, which typically investigates comparably high doses of drug over a much shorter timeframe. The MR result may therefore not directly correspond to the effect size in practice and does not perfectly predict the effect of a drug.

In addition, the eQTL cohorts contained some non-European individuals[17,18], three of the pQTL studies sourced were based on Icelandic, Scottish and German cohorts[38–40], and the PD populations were comprised of European individuals only[9–11]. Linkage disequilibrium patterns differ between populations, which may compromise how well our QTLs mimic drug action in the PD cohorts and introduce bias to the MR effect estimate[25].

It is difficult to interpret which tissue would be the most appropriate site of action. Whereas the genes that reached significance in both blood and brain tissue may have stronger MR evidence, targeting the protein of a widely expressed gene may lead to systemic side-effects. Brain tissue may be more biologically relevant for neurodegeneration, but a drug acting in the blood stream may not need to cross the blood–brain barrier to exert its effect. A limitation of using brain tissue is that gene expression is quantified post-mortem, and measured expression levels are influenced by RNA degradation occurring after cell death as well as transcriptional changes occurring in response to death[55]. We included both blood and brain tissue eQTLs to capture as many genes as possible and explore two potential tissue sites of action, but we note that it is difficult to prioritise genes based on which tissue(s) they reached significance in.

Furthermore, the sample size of our blood eQTL data ($n = 31,684$) is larger than that for brain tissue eQTLs ($n = 1387$) and the blood pQTL study ($n = 750–4137$). A larger sample size allows greater power to detect QTLs, meaning there are more SNPs per gene. Nevertheless, it is unclear how well QTL data mimic medications that modulate activity levels of the protein. We are encouraged that five of the seven proteins we were able to probe using both eQTL and pQTL data were successfully validated, adding to existing evidence that regulatory variants may be used for robust causal inferences in drug target MR[8]. Nevertheless, this MR study does not provide functional evidence for the proposed drug targets, and the MR process does not replace pre-clinical evaluation of drug targets in vitro and in vivo. Genomic approaches serve as adjuncts thereto, promising to better prioritise drug targets carried forward to clinical phase trials.

A 9.6% vs. 13.8% success rate for drugs from phase 1 trials to approval may mean a $480 million difference in the median research and development cost of bringing a new drug to the market[1]. The druggable genome resource has opened up new avenues for drug target identification using existing genetic data[7,56,57], and if genetic evidence increases success rates even by a few percent, this could have a substantial effect on drug development costs[4,5]. As such, MR a highly compelling, time- and cost-effective adjunct to the randomized controlled trial. We have made our code openly available for use beyond PD research (https://github.com/catherinestorm/mr_druggable_genome_pd/)[58], and we have demonstrated ways to prioritise drug targets based on genetic data. We have provided human genetic evidence of drug efficacy for PD, and we hope that these data will serve as a useful resource for prioritising drug development efforts.

## Methods
All DNA positions are based on the human reference genome build hg19 (GRCh37). Data processing was completed using R software version 3.6.3[59].

**Exposure data**. Tissue-specific eQTL data were obtained from the eQTLGen (https://eqtlgen.org/) and PsychENCODE consortia (http://resource.psychencode.org/); full descriptions of the data are available in the original publications[17,18]. Briefly, the eQTLGen data consisted of cis-eQTLs for 16,987 genes and 31,684 blood samples, of which most were healthy European-ancestry individuals. We downloaded the full significant cis-eQTL results (FDR < 0.05) and allele frequency information from the eQTLGen consortium on 13 May 2020.

The PsychENCODE data included 1387 prefrontal cortex, primarily-European samples (679 healthy controls, 497 schizophrenia, 172 bipolar disorder, 31 autism spectrum disorder and 8 affective disorder patients). We downloaded all significant eQTLs (FDR < 0.05) for genes with expression >0.1 fragments per kilobase per million mapped fragments (FPKM) in at least ten samples and all SNP information, accessed on 13 May 2020.

We obtained an updated version of the druggable genome containing 4863 genes from the authors of the original publication[7], double-checking the druggability level for all genes marked as approved or in clinical trials ("druggability tier 1"). We removed non-autosomal genes, leaving 4560 druggable genes. We filtered both eQTL datasets to include SNPs 5 kb upstream of the target druggable gene start or 5 kb downstream of the target druggable gene end position.

We sought freely available pQTL data from blood or brain tissue for all druggable genes that reached significance for any PD outcome in our study. Out of 23 pQTL studies identified, four studies (1) reported significant pQTLs in individuals of European descent for any of the druggable proteins proposed by our eQTL analysis, (2) provided all the SNP information required for MR and (3) reported SNPs that were available in our PD outcome data[37–40]. Sun and colleagues measured 3622 proteins in 3301 healthy European blood donors from the INTERVAL study and identified 1927 pQTLs for 1478 proteins. Emilsson and colleagues measured 4137 proteins in the serum of 5457 Icelanders from AGES Reykjavik study. Effect alleles and effect allele frequencies were obtained from the authors. Suhre and colleagues measured 1124 proteins in 1000 blood samples from a German population. Hillary and colleagues measured 92 proteins in the blood of 750 healthy Scottish controls.

In total, we found pQTLs that were available in the appropriate PD outcome data for seven of our druggable proteins of interest: BST1, CD38, CTSB, GPNMB, LGALS3, PYGL and QDPR. All pQTLs included in our analysis had $p < 5e-6$ in the original pQTL study. All pQTLs were found on the same chromosome as the associated gene except for: rs62143198 for PYGL, rs62143197 for QDPR, rs4253282 for GPNMB, rs2731674 for GPNMB[37]. These latter four SNPs are therefore acting in trans.

**Outcome data.** All PD data were obtained from the IPDGC, and details on recruitment and quality control are available in the original publications[9–11,21]. In the discovery phase for PD risk, we used openly available summary statistics from a 2014 case-control GWAS meta-analysis, which included 13,708 PD patients and 95,282 controls[21].

In the replication phase for PD risk, we obtained summary statistics from 11 case-control GWAS studies included in the most recent PD risk GWAS meta-analysis from the authors[9]. The 11 studies, as named and described in the PD GWAS meta-analysis, were Spanish Parkinson's, Baylor College of Medicine/University of Maryland, McGill Parkinson's, Oslo Parkinson's Disease Study, Parkinson's Progression Markers Initiative (PPMI), Finnish Parkinson's, Harvard Biomarker Study (HBS), UK PDMED (CouragePD), Parkinson's Disease Biomarker's Program (PDBP), Tübingen Parkinson's Disease cohort (CouragePD) and Vance (dbGap phs000394). These yielded a total of 8036 PD cases and 5803 controls. We meta-analysed the data using METAL (version 2011-03-25) using default settings, weighting by sample size[60]. The overall genomic inflation factor was $\lambda = 1.116$, and when scaled to 1000 cases and 1000 controls $\lambda_{1000} = 1.017$. Based on genomic inflation factors and quantile–quantile plots of the original GWASs[9,21], we considered our quantile–quantile plot to show adequate agreement with the expected null distribution (Supplementary Fig. 1).

For the progression marker analyses, we used summary statistics from the largest publicly available GWAS meta-analyses for PD age at onset and clinical progression[10,11]. For age at onset, this inceleed 17,996 PD cases, and age at onset was defined as self-reported age at motor symptom onset or PD diagnosis. The authors reported a high correlation between age at diagnosis and age at onset.

The progression GWAS meta-analysis included 4093 PD patients from 12 cohorts, followed over a median of 2.97 years (mean visits per individual over the study period: 5.44). We downloaded summary statistics for nine continuous outcomes and four binomial outcomes (https://pdgenetics.shinyapps.io/pdprogmetagwasbrowser/). Continuous outcomes included Hoehn and Yahr stage (PD progression rating scale), total UPDRS/Movement Disorder Society revised version total (PD progression rating scale), UPDRS parts 1 to 4 (1 = non-motor symptoms, 2 = motor symptoms, 3 = motor examination, 4 = motor complications), MOCA (cognitive impairment), MMSE (cognitive impairment) and SEADL (activities of daily living and independence). The binomial outcomes we used were dementia, depression, dyskinesia, as well as reaching Hoehn and Yahr stage 3 or more.

**Mendelian randomization.** MR analyses were completed using the R package "TwoSampleMR" (version 0.5.4)[61], unless stated otherwise. The exposure and outcome data were loaded and harmonized using in-built functions. SNPs were then clumped at $r^2 < 0.2$ using European samples from the 1000 Genomes Project[20,61]. Steiger filtering was used to remove genes where SNPs explained a greater proportion of variation in the outcome (PD trait) than variation in the exposure (gene expression). For the eQTL analysis, the Steiger filtering excluded 0–403 genes per outcome tested in a tissue, representing 0–15% of all genes studied per outcome tested in a tissue.

Wald ratios were calculated for all SNPs. These were meta-analysed using the IVW, MR-Egger and maximum likelihood methods, including a linkage disequilibrium matrix to account for correlation between SNPs; this function uses the R package "MendelianRandomization" version 0.4.2[29]. Forest plots were produced using the R package "forestplot".

Where >2 SNPs were available per exposure, we assessed whether the MR-Egger intercept significantly deviated from zero, as well as Cochran's $Q$ and $I^2$ methods to test for heterogeneity between Wald ratios[62]. FDR-corrected p-values were calculated within each exposure-outcome combination to correct for multiple testing. In the discovery study for PD risk and the PD progression studies, we considered FDR < 0.05 significant. In the replication studies for PD risk and age at onset, as well as the pQTL study, we considered nominal $p < 0.05$ significant.

For genes which reached significance using the IVW method (>1 SNP available), we carried out another MR analysis, clumping at $r^2 < 0.001$. If >1–2 SNPs were available at this clumping threshold, Wald ratios were meta-analysed using the IVW, MR-Egger, weighted mode and weighted median methods.

**Colocalization.** We carried out a colocalization analysis for PD risk, age at onset and progression outcomes using the R package "coloc"[31]. We harmonized exposure and outcome datasets using the "TwoSampleMR" package. We used default priors: $p1 = 10^{-4}$, $p2 = 10^{-4}$, $p12 = 10^{-5}$. p1, p2 and p12 are the prior probabilities that a SNP in the tested region is significantly associated with expression of the tested gene, the tested PD outcome, or both, respectively. The colocalization yields

posterior probabilities corresponding to one of five hypotheses: PPH0, no association with either trait; PPH1, association with expression of the gene, but not the PD trait; PPH2, association with the PD trait, but not expression of the gene; PPH3, association with the PD trait and expression of the gene, with distinct causal variants; PPH4, association with the PD trait and expression of the gene, with a shared causal variant[31]. A low PPH3 and PPH4 in combination with a high PPH0, PPH1 and/or PPH2 indicates limited power in the colocalization analysis[31]. We therefore restricted our analysis to genes reaching $PPH3 + PPH4 \geq 0.8$.

**Reporting summary.** Further information on research design is available in the Nature Research Reporting Summary linked to this article.

## Data availability
The data generated in this study and a data dictionary (Supplementary Information) are provided in the Supplementary Information. Tissue-specific eQTL data were obtained from the eQTLGen (https://eqtlgen.org/) and PsychENCODE consortia (http://resource.psychencode.org). The druggable genome data were obtained via from the original authors; an immediately accessible version of the druggable genome is available in the supplementary materials of the original publication[7]. Subsets of the most recent PD risk GWAS were obtained from the original authors[9]. GWAS summary statistics for PD age at onset are available on the IPDGC website (http://pdgenetics.org/resources). Parkinson's progression GWAS data can be found here: https://pdgenetics.shinyapps.io/pdprogmetagwasbrowser/.

## Code availability
The code used for this study is openly available on GitHub, accompanied by instructions for use and required tools (https://github.com/catherinestorm/mr_druggable_genome_pd). Data processing was completed using R software (version 3.6.3), with packages TwoSampleMR (version 0.5.4), MendelianRandomization (version 0.4.2), coloc (version 4.0.4), dplyr (version 1.0.0), readr (version 1.3.1), stringr (version 1.4.0), tidyverse (version 1.3.0), forestplot (version 1.9), plyr (version 1.8.6), devtools (version 2.3.0), remotes (version 2.1.1). The GWAS meta-analysis was completed using METAL (version 2011-03-25).

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

## Acknowledgements

C.S.S. would like to thank Dr. Vishal Rawji for his invaluable support and insightful ideas about the clinical implications and communication of this study. C.S.S. is funded by Rosetrees Trust, John Black Charitable Foundation and the University College London MBPhD Programme. D.A.K. is supported by an MBPhD Award from the International Journal of Experimental Pathology. M.A. is funded by the Faculty of Applied Medical Sciences, King Abdulaziz University, Jeddah, Saudi Arabia. N.W.W. and A.D.H. are National Institute for Health Research senior investigators. N.W.W., A.D.H. and C.F. receive support from the National Institute for Health Research University College London Hospitals Biomedical Research Centre. We would like to thank all members of the International Parkinson Disease Genomics Consortium (IPDGC) and the authors of QTL projects referenced here, who make their data openly available. We thank all the patients and families whose decision to donate tissue samples make our research possible.

## Author contributions

Conceptualization: C.S.S., D.A.K., M.M.A. and N.W.W.; methodology: C.S.S., D.A.K., M.M.A. and N.W.W.; investigation, formal analysis, visualization: C.S.S.; resources: S.B.C., C.F., A.D.H. and IPDGC; writing—original draft: CSS; writing—review & editing: C.S.S., D.A.K., M.M.A., N.W.W., S.B.C., C.F., A.D.H. and IPDGC.

## Competing interests

The authors declare no competing interests.

**Additional information**

## International Parkinson's Disease Genomics Consortium (IPDGC)

Alastair J. Noyce[8,9], Rauan Kaiyrzhanov[10], Ben Middlehurst[11], Demis A. Kia[1], Manuela Tan[12], Henry Houlden[10], Catherine S. Storm[1], Huw R. Morris[12], Helene Plun-Favreau[10], Peter Holmans[13,14], John Hardy[10], Daniah Trabzuni[10], John Quinn[11], Vivien Bubb[11], Kin Y. Mok[10], Kerri J. Kinghorn[15], Nicholas W. Wood[1✉], Patrick Lewis[16], Sebastian R. Schreglmann[10], Ruth Lovering[17], Lea R'Bibo[10], Claudia Manzoni[16], Mie Rizig[10], Mina Ryten[10], Sebastian Guelfi[10], Valentina Escott-Price[18], Viorica Chelban[10], Thomas Foltynie[19], Nigel Williams[20], Karen E. Morrison[21], Carl Clarke[22,23], Kirsten Harvey[24], Benjamin M. Jacobs[8], Alexis Brice[25], Fabrice Danjou[25], Suzanne Lesage[25], Jean-Christophe Corvol[26], Maria Martinez[27,28], Claudia Schulte[29,30], Kathrin Brockmann[29,30], Javier Simón-Sánchez[29,30], Peter Heutink[31], Patrizia Rizzu[32], Manu Sharma[33], Thomas Gasser[29,30], Susanne A. Schneider[34], Mark R. Cookson[3], Sara Bandres-Ciga[3], Cornelis Blauwendraat[3], David W. Craig[35], Kimberley Billingsley[3], Mary B. Makarious[3], Derek P. Narendra[36], Faraz Faghri[3,37], J. Raphael Gibbs[38], Dena G. Hernandez[3], Kendall Van Keuren-Jensen[39], Joshua M. Shulman[40,41], Hirotaka Iwaki[3], Hampton L. Leonard[3], Mike A. Nalls[3,42], Laurie Robak[43], Jose Bras[44], Rita Guerreiro[44], Steven Lubbe[45], Timothy Troycoco[46], Steven Finkbeiner[47,48,49], Niccolo E. Mencacci[50], Codrin Lungu[51], Andrew B. Singleton[3], Sonja W. Scholz[52], Xylena Reed[3], Ryan J. Uitti[53], Owen A. Ross[54], Francis P. Grenn[3], Anni Moore[3], Roy N. Alcalay[55,56], Zbigniew K. Wszolek[53], Ziv Gan-Or[57], Guy A. Rouleau[57], Lynne Krohn[57], Kheireddin Mufti[57], Jacobus J. van Hilten[58], Johan Marinus[58], Astrid D. Adarmes-Gómez[59], Miquel Aguilar[60], Ignacio Alvarez[60], Victoria Alvarez[61], Francisco Javier Barrero[62], Jesús Alberto Bergareche Yarza[63], Inmaculada Bernal-Bernal[59], Marta Blazquez[61], Marta Bonilla-Toribio[59], Juan A. Botía[64], María Teresa Boungiorno[60], Dolores Buiza-Rueda[59], Ana Cámara[65], Fátima Carrillo[59], Mario Carrión-Claro[59], Debora Cerdan[66], Jordi Clarimón[67,68], Yaroslau Compta[65], Monica Diez-Fairen[60], Oriol Dols-Icardo[67,68], Jacinto Duarte[66], Raquel Duran[69], Francisco Escamilla-Sevilla[70], Mario Ezquerra[65], Cici Feliz[71], Manel Fernández[65], Rubén Fernández-Santiago[65], Ciara Garcia[61], Pedro García-Ruiz[72], Pilar Gómez-Garre[59], Maria Jose Gomez Heredia[73], Isabel Gonzalez-Aramburu[74], Ana Gorostidi Pagola[63], Janet Hoenicka[75], Jon Infante[68,76], Silvia Jesús[59], Adriano Jimenez-Escrig[77], Jaime Kulisevsky[68,78], Miguel A. Labrador-Espinosa[59], Jose Luis Lopez-Sendon[77], Adolfo López de Munain Arregui[63], Daniel Macias[59], Irene Martínez Torres[79], Juan Marín[68,78], Maria Jose Marti[80], Juan Carlos Martínez-Castrillo[81], Carlota Méndez-del-Barrio[59], Manuel Menéndez González[61], Marina Mata[82], Adolfo Mínguez[82], Pablo Mir[59], Elisabet Mondragon Rezola[63], Esteban Muñoz[80], Javier Pagonabarraga[78,83], Pau Pastor[60], Francisco Perez Errazquin[73], Teresa Periñán-Tocino[59], Javier Ruiz-Martínez[84], Clara Ruz[69], Antonio Sanchez Rodriguez[74], María Sierra[74], Esther Suarez-Sanmartin[61], Cesar Tabernero[66],

Juan Pablo Tartari[60], Cristina Tejera-Parrado[59], Eduard Tolosa[80], Francesc Valldeoriola[80], Laura Vargas-González[59], Lydia Vela[85], Francisco Vives[69], Alexander Zimprich[86], Lasse Pihlstrom[87], Mathias Toft[88], Pille Taba[89], Sulev Koks[90,91], Sharon Hassin-Baer[92,93], Kari Majamaa[94,95], Ari Siitonen[94,95], Pentti Tienari[96,97], Njideka U. Okubadejo[98], Oluwadamilola O. Ojo[98], Chingiz Shashkin[99], Nazira Zharkinbekova[100], Vadim Akhmetzhanov[101], Gulnaz Kaishybayeva[102], Altynay Karimova[102], Talgat Khaibullin[103] & Timothy L. Lynch[104,105]

[8]Preventive Neurology Unit, Wolfson Institute of Preventive Medicine, QMUL, London, UK. [9]Department of Molecular Neuroscience, UCL, London, UK. [10]Department of Molecular Neuroscience, UCL Institute of Neurology, London, UK. [11]Institute of Translational Medicine, University of Liverpool, Liverpool, UK. [12]Department of Clinical Neuroscience, University College London, London, UK. [13]Biostatistics and Bioinformatics Unit, Institute of Psychological Medicine and Clinical Neuroscience, Cardiff, UK. [14]MRC Centre for Neuropsychiatric Genetics & Genomics, Cardiff, UK. [15]Institute of Healthy Ageing, Research Department of Genetics, Evolution and Environment, University College London, London, UK. [16]University of Reading, Reading, UK. [17]University College London, London, UK. [18]MRC Centre for Neuropsychiatric Genetics and Genomics, Cardiff University School of Medicine, Cardiff, UK. [19]UCL Institute of Neurology, London, UK. [20]MRC Centre for Neuropsychiatric Genetics and Genomics, Cardiff, UK. [21]Faculty of Medicine, University of Southampton, Southampton, UK. [22]University of Birmingham, Birmingham, UK. [23]Sandwell and West Birmingham Hospitals NHS Trust, Birmingham, UK. [24]UCL School of Pharmacy, London, UK. [25]Institut du Cerveau et de la Moelle épinière, ICM, Inserm U 1127, CNRS, UMR 7225, Sorbonne Universités, UPMC University Paris 06, UMR S 1127, AP-HP, Pitié-Salpêtrière Hospital, Paris, France. [26]Institut du Cerveau et de la Moelle épinière, ICM, Inserm U 1127, CNRS, UMR 7225, Sorbonne Universités, UPMC University Paris 06, UMR S 1127, Centre d'Investigation Clinique Pitié Neurosciences CIC-1422, AP-HP, Pitié-Salpêtrière Hospital, Paris, France. [27]INSERM UMR 1220, Toulouse, France. [28]Paul Sabatier University, Toulouse, France. [29]Department for Neurodegenerative Diseases, Hertie Institute for Clinical Brain Research, University of Tübingen, Tübingen, Germany. [30]DZNE, German Center for Neurodegenerative Diseases, Tübingen, Germany. [31]DZNE, German Center for Neurodegenerative Diseases and Department for Neurodegenerative Diseases, Hertie Institute for Clinical Brain Research, University of Tübingen, Tübingen, Germany. [32]DZNE, German Center for Neurodegenerative Diseases, Bonn, Germany. [33]Centre for Genetic Epidemiology, Institute for Clinical Epidemiology and Applied Biometry, University of Tubingen, Tübingen, Germany. [34]Department of Neurology, Ludwig-Maximilians-University Munich, München, Germany. [35]Department of Translational Genomics, Keck School of Medicine, University of Southern California, Los Angeles, CA, USA. [36]Inherited Movement Disorders Unit, National Institute of Neurological Disorders and Stroke, Bethesda, MD, USA. [37]Department of Computer Science, University of Illinois at Urbana-Champaign, Urbana, IL, USA. [38]Laboratory of Neurogenetics, National Institute on Aging, National Institutes of Health, Bethesda, MD, USA. [39]Neurogenomics Division, TGen, Phoenix, AZ, USA. [40]Departments of Neurology, Neuroscience, and Molecular & Human Genetics, Baylor College of Medicine, Houston, TX, USA. [41]Jan and Dan Duncan Neurological Research Institute, Texas Children's Hospital, Houston, TX, USA. [42]CEO/Consultant Data Tecnica International, Glen Echo, MD, USA. [43]Baylor College of Medicine, Houston, TX, USA. [44]Center for Neurodegenerative Science, Van Andel Research Institute, Grand Rapids, MI, USA. [45]Ken and Ruth Davee Department of Neurology and Simpson Querrey Center for Neurogenetics, Northwestern University Feinberg School of Medicine, Chicago, IL, USA. [46]National Institutes of Health, Bethesda, MD, USA. [47]Departments of Neurology and Physiology, University of California, San Francisco, CA, USA. [48]Gladstone Institute of Neurological Disease, San Francisco, CA, USA. [49]Taube/Koret Center for Neurodegenerative Disease Research, San Francisco, CA, USA. [50]Northwestern University Feinberg School of Medicine, Chicago, IL, USA. [51]National Institutes of Health Division of Clinical Research, NINDS, National Institutes of Health, Bethesda, MD, USA. [52]Neurodegenerative Diseases Research Unit, National Institute of Neurological Disorders and Stroke, Bethesda, MD, USA. [53]Department of Neurology, Mayo Clinic, Jacksonville, FL, USA. [54]Departments of Neuroscience & Clinical Genomics, Mayo Clinic, Jacksonville, FL, USA. [55]Department of Neurology, College of Physicians and Surgeons, Columbia University Medical Center, Taub Institute for Research on Alzheimer's Disease, New York, NY, USA. [56]The Aging Brain, College of Physicians and Surgeons, Columbia University Medical Center, New York, NY, USA. [57]Montreal Neurological Institute and Hospital, Department of Neurology & Neurosurgery, Department of Human Genetics, McGill University, Montréal, QC H3A 0G4, Canada. [58]Department of Neurology, Leiden University Medical Center, Leiden, Netherlands. [59]Instituto de Biomedicina de Sevilla IBiS Hospital Universitario Virgen del Rocío/CSIC/Universidad de Sevilla, Seville, Spain. [60]Fundació Docència i Recerca Mútua de Terrassa and Movement Disorders Unit, Department of Neurology, University Hospital Mutua de Terrassa, Terrassa, Barcelona, Spain. [61]Hospital Universitario Central de Asturias, Oviedo, Spain. [62]Hospital Universitario San Cecilio de Granada, Universidad de Granada, Granada, Spain. [63]Instituto de Investigación Sanitaria Biodonostia, San Sebastián, Spain. [64]Universidad de Murcia, Murcia, Spain. [65]Hospital Clinic de Barcelona, Barcelona, Spain. [66]Hospital General de Segovia, Segovia, Spain. [67]Memory Unit, Department of Neurology, IIB Sant Pau, Hospital de la Santa Creu i Sant Pau, Universitat Autònoma de Barcelona, Barcelona, Spain. [68]Centro de Investigación Biomédica en Red en Enfermedades Neurodegenerativas (CIBERNED), Madrid, Spain. [69]Centro de Investigacion Biomedica, Universidad de Granada, Granada, Spain. [70]Hospital Universitario Virgen de las Nieves, Instituto de Investigación Biosanitaria de Granada, Granada, Spain. [71]Departmento de Neurologia, Instituto de Investigación Sanitaria Fundación Jiménez Díaz, Madrid, Spain. [72]Instituto de Investigación Sanitaria Fundación Jiménez Díaz, Madrid, Spain. [73]Hospital Universitario Virgen de la Victoria, Malaga, Spain. [74]Hospital Universitario Marqués de Valdecilla-IDIVAL, Santander, Spain. [75]Institut de Recerca Sant Joan de Déu, Barcelona, Spain. [76]Hospital Universitario Marqués de Valdecilla-IDIVAL and University of Cantabria, Santander, Spain. [77]Hospital Universitario Ramón y Cajal, Madrid, Spain. [78]Movement Disorders Unit, Department of Neurology, IIB Sant Pau, Hospital de la Santa Creu i Sant Pau, Universitat Autònoma de Barcelona, Barcelona, Spain. [79]Department of Neurology, Instituto de Investigación Sanitaria La Fe, Hospital Universitario y Politécnico La Fe, Valencia, Spain. [80]Hospital Clinic Barcelona, Barcelona, Spain. [81]Instituto Ramón y Cajal de Investigación Sanitaria, Hospital Universitario Ramón y Cajal, Madrid, Spain. [82]Hospital Universitario Virgen de las Nieves, Granada, Instituto de Investigación Biosanitaria de, Granada, Spain. [83]Department of Neurology, Hospital Universitario Infanta Sofía, Madrid, Spain. [84]Hospital Universitario Donostia, Instituto de Investigación Sanitaria Biodonostia, San Sebastián, Spain. [85]Department of Neurology, Hospital Universitario Fundación Alcorcón, Madrid, Spain. [86]Department of Neurology, Medical University of Vienna, Vienna, Austria. [87]Department of Neurology, Oslo University Hospital, Oslo, Norway. [88]Department of Neurology and Institute of Clinical Medicine, Oslo University Hospital, Oslo, Norway. [89]Department of Neurology and Neurosurgery, University of Tartu, Tartu, Estonia. [90]Centre for Molecular Medicine and Innovative Therapeutics, Murdoch University, Murdoch, 6150 Perth, Western Australia, Australia. [91]The Perron Institute for Neurological and Translational Science, Nedlands, 6009 Perth, Western Australia, Australia. [92]The Movement Disorders Institute, Department of Neurology and Sagol Neuroscience Center, Chaim Sheba Medical Center, Tel-Hashomer, 5262101 Ramat

Gan, Israel. [93]Sackler Faculty of Medicine, Tel Aviv University, Tel Aviv, Israel. [94]Institute of Clinical Medicine, Department of Neurology, University of Oulu, Oulu, Finland. [95]Department of Neurology and Medical Research Center, Oulu University Hospital, Oulu, Finland. [96]Clinical Neurosciences, Neurology, University of Helsinki, Helsinki, Finland. [97]Helsinki University Hospital, Helsinki, Finland. [98]University of Lagos, Lagos State, Nigeria. [99]Kazakh National Medical University named after Asfendiyarov, Almaty, Kazakhstan. [100]South Kazakhstan Medical Academy, Shymkent, Kazakhstan. [101]Astana Medical University, Astana, Kazakhstan. [102]Scientific and Practical Center "Institute of Neurology named after Smagul Kaishibayev", Almaty, Kazakhstan. [103]Semey Medical University, Semey, Kazakhstan. [104]School of Medicine and Medical Science, University College Dublin, Dublin, Ireland. [105]The Dublin Neurological Institute at the Mater Misericordiae University Hospital, Dublin, Ireland.

