## [Peer Review File · Nature Communications]

Finding genetically-supported drug targets for Parkinson's disease using Mendelian randomization of the druggable genomeReviewers' Comments:

Reviewer #1:

Remarks to the Author:

This paper provides genomic evidence for a range of potential drug targets for Parkinson's disease. The results are novel and will be of great interest to the wider field. This method - drug target MR - (which has been developed and promoted by the authors), has the potential to provide more reliable evidence of drug target efficacy than other approaches. As far as I am aware, there is no other paper doing comparable analysis for PD. The authors' conclusions are supported by their results and analysis, and they do a great job of providing detailed code etc. I don't see any substantial flaws in their analysis or interpretation, certainly none that preclude publication. I've made some suggestions below that may improve the paper - but these just relate to the clarity of the paper rather than scientific validity. The paper definitely meets the standards expected of the field and the authors provide a detailed GitHub repo of all the code used in the analysis - so the work is likely to be reproducible (note I have not attempted to reproduce their results using their code).

How many of the eQTL and pQTLs are cis versus trans SNPs? In general, for drug target MR, the number of SNPs tends to be far lower than for other forms of MR. This is not an issue, but often the focus is on cis-QTLs which are more proximal to the gene of interest, and are likely to have fewer pleiotropic effects. Furthermore, MR-Egger other sensitivity analyses are more rarely useful because the number of SNPs and the variation in the instrument-exposure association can be very low, leading to very low power. I don't think major changes are needed for this, but it might be good to note the number and proportion of cis versus trans SNPs you're using.

MR studies of disease progression can suffer from collider bias, in which factor which affects the incidence of disease can appear spuriously associated with progression in MR studies. How did you handle this, and can you be certain that your results for disease progression variants are not due to collider bias? Note - you should be able to address this just in the discussion - for example, by noting the difference in factors and genes identified for incidence and progression makes this explanation unlikely. It may also be worth flagging this as a potential limitation in the discussion "for future research", as there are methods that are/have been developed for this issue.

It's great you've a GitHub repo for this. Given all the data are summary data, it might be worth uploading this to GitHub as well. But again, totally up to you.

How many SNPs were excluded due to Stieger filtering?

Neil Davies

1 Paternoster L, Tilling K, Davey Smith G. Genetic epidemiology and Mendelian randomization for informing disease therapeutics: Conceptual and methodological challenges. *PLOS Genetics* 2017;13:e1006944. doi:10.1371/journal.pgen.1006944

2 Munafò MR, Tilling K, Taylor AE, et al. Collider scope: when selection bias can substantially influence observed associations. *Int J Epidemiol* 2018;47:226–35. doi:10.1093/ije/dyx206

Reviewer #2:

Remarks to the Author:

In this study, the authors utilize public GWAS data on Parkinson's disease (PD) risk and severity together with public eQTL and pQTL data from blood and brain to conduct two-sample Mendelian Randomization (MR) analyses with the aim to prioritize new drug targets for PD. Starting from a set of ~3,000 "druggable" proteins they suggest 23 genes with causal links to PD, incl. four which they

propose as suitable for drug repositioning. The study aims to address a substantial need (identifying drug targets for PD with a high likelihood of success) using a very promising approach (MR in its multiple forms), yet it falls short to validate the results with the rigor needed to propose new drug targets so that overall the insights gained from this manuscript beyond the existing literature are modest.

1. The authors agree (2nd sentence in discussion) that formal colocalization evidence is needed to reliably call drug targets from MR, but they have not done this in this study (apart from citing colocalization evidence for CD38 and GPNMB from earlier pQTL studies). How many targets are still left from the broader candidate set when an appropriate colocalization cut-off (e.g. $R^2 > 0.7$) is being used?

2. Pleiotropy is disregarded in the decision what constitutes a potential drug target. The manuscript only tests a selected set of instruments on "druggable" PD genes, but is not thoroughly investigating what effect the eQTLs and pQTLs may have on the expression of other genes nearby or in trans (be they part of the "druggable" genome or not).

3. The paper only concentrates on efficacy signals for PD but does not consider safety aspects, alternative indications or indirect mechanisms which also need to be considered when proposing and validating the suitability of a gene as drug target. For instance a quick look-up in the Open Targets genetics platform indicates that the GPNMB trans-pQTL rs4253282 near the F11 gene might also be associated with a range of blood clotting and vascular endpoints that may independently modulate PD risk (instead of impacting PD risk through modulating GPNMB levels). PheWAS data for these types of analyses are publicly available, e.g. from GWAS catalog, UK Biobank or FinnGen and a critical component of the target evaluation process (if the main emphasis of the paper is on nominating drug targets as it is currently being pitched).

4. MR methods accounting for LD between instruments should be employed rather than relying on LD with an arbitrary/lenient R^2 of 0.2 or distance-based cut-offs.

5. It is not clear to me how exactly multiple testing corrections have been performed throughout the analyses. FDR is being mentioned, but has this been applied to risk or progression markers or both, and genome-wide or just for the candidate set. Please elaborate.

6. By how much does clumping of just 2-3 instruments really improve MR estimates? Does it improve the accuracy at all and justify why this authors focus their attention on genes supported by >1 eQTL?

7. Psychencode is single source of brain eQTLs in this manuscripts. While laudable that eQTLs are being tested from a tissue probably most relevant to PD, the small sample size may challenge e.g. directionality estimates which are critical for choosing the therapeutic approach. For instance, a claim that the HSD3B7 shows a different directionality in brain and blood should be supported by independent data. Also, the caveats when relying on (probably in most cases post-mortem) brain expression data for eQTL calling should be discussed.

8. Related to point 7., repurposing of Roledumab targeting RHD is supported by brain eQTL but apparently not blood eQTL data (or I have overlooked). Roledumab acts in the blood, not the brain. Why should this be a good repurposing opportunity for PD?

9. p.3 "SNPs associated with expression levels of a gene (expression quantitative trait loci, eQTLs) are analogous to lifelong exposure to a medication targeting the encoded protein". In this context, "are" should be substituted by "may be", but this sentence is exemplary for the remainder of the manuscript that needs to be carefully scrutinized for overstatements.

10. Throughout the manuscript, the limitations of MR to predict drug targets are insufficiently

highlighted. In addition to highlighting in the discussion general challenges (like lifelong genetic modulation vs acute therapeutic modulation, Europeans vs non-Europeans, tissue,...) the authors need to detail better what assumptions the respective MR methods used are making and how easily they can be violated. For instance, challenges of utilizing weak instruments, the volatility of directionality based on messy gene expression data forming the foundation for eqTL calling, or that the number of instruments available for clumping methods is typically very modest,... may lead to faulty assumptions is not discussed. Overall, the broad claims in the manuscript are often not substantiated by sufficiently robust data.

Reviewer #3:

Remarks to the Author:

Finding drug-targeting mechanisms with genetic evidence for Parkinson's disease _Mendelian randomization of the druggable genome

Catherine S. Storm¹, Demis A. Kia¹, Mona Almramhi^{1,2}, Sara Bandres-Ciga², Chris Finan⁴, Aron D. Hingorani^{4,5,6}, IPDGC +, Nicholas W. Wood^{1*}

Mendelian Randomization of druggable genome is a clever idea that has been gaining attention. We have a major problem with drug development for neurodegenerative diseases: staggering failure in clinical trials, much disappointment, extended suffering of people with disease, and fortunes spent. Using genetics to inform drug targets is of course a great idea, especially considering nearly all the work can be done fast and cheap in silico, using the online databases of genotypes, gene expression, proteins, and druggable genes.

The approach has been published and applied to other disorders: ex.

<https://pubmed.ncbi.nlm.nih.gov/33005893/>; <https://www.nature.com/articles/s41380-019-0540-z>. Here, authors try it on Parkinson's disease for the first time.

Among their findings, authors nominate NDUFAF2/metformin as the one that is best suited for repurposing for PD. I appreciate authors' open discussion on how their other findings may not be good ideas for PD. I agree with their summation, although their data alone is not unequivocal. Added support comes from published data on association of type 2 diabetes with PD, and on the association of metformin use with lower incidence of PD. As added relevance, see <https://www.pnas.org/content/117/42/26438> which came out in Oct 2020 while the present paper was in review.

NDUFAF2 did not pass Cochran's Q nor I2 test. This statement is made in the context of pleiotropy. Please explain what this finding means and how it should be interpreted (I know what Q and I2 are used for, and what pleiotropy is, but don't understand how the former can inform on the latter).

The usual use of Cochran's Q and I2 is to test heterogeneity. I cannot find in methods how these tests were applied here. If in the usual context, then it would imply NDUFAF2/metformin result is highly heterogeneous (per statistics in supplement). Does that mean metformin may be a useful preventative drug for a subset of PD? That begs the question: which subset? This paper does not address the heterogeneity in PD. I suggest they do, and discuss how their analyses deal with the unaccounted heterogeneity in PD.

It is not surprising that there is little overlap between risk and progression, as highlighted as surprising even in abstract. They point to different things: risk points to an initial trigger (an external insult or a rare pathogenic mutation), progression points to intrinsic mechanisms that determine what happens in response to that insult. I think the more interesting question is the overlap between age at onset and progression, since the former can be viewed as the early phase of the latter. This study did not (could not with current study design) answer the overlap between age at onset and progression,

but if they are amenable to additional analyses, they have the data and can attempt it.

Authors provide effect sizes: OR for risk, beta for age at onset and progression. How should these values be interpreted in the context of druggability? Can one infer drug efficacy from these effect sizes, if so, how?

As authors point out, the drugs for progression would be of more immediate interest than drugs for prevention, because we cannot predict PD and don't know who should be subject to prevention. (metformin is with risk/prevention). The analysis for progression is limited by short duration of follow-up (median ~3 years) and not having a second dataset for replication. I suggest authors take advantage of other publicly available datasets to improve their analysis of progression. PPMI for example is longitudinal, started with de novo unmedicated PD patients, and has followed them since 2013.

There are many caveats that call for "caution". Authors acknowledge this. They present the criteria they use to call a result more robust than others – they are subjective criteria. I disagree with using association with risk and age at onset as corroborating evidence of association with two phenotypes – in reality they are distinct, but the way we measure them and treat them statistically, they can mimic each other (association with higher risk often also presents as earlier onset, and protection with delayed onset).

QQ plot (Figure S1) shows alarming inflation. It is said to have "adequate agreement with the expected null distribution". Can you explain why it looks so inflated when inflation factor is 1.1?

Need data dictionary for supplemental tables. Supplementary material can use a thorough cleaning-up (ex. Nalls et al is not a phenotype).

In summary, the method is clever. The application to PD is new. The paper doesn't provide any conclusive results on PD (they can still try), but it does demonstrate how genomics resources can be used to inform clinical trials. Just have to get the attention of people who conduct clinical trials. Should the authors decide to address the translation to clinical practice, they should add a paragraph or two on the steps that will have to be taken from identifying a candidate drug in silico to using it on people, maybe with an example for prevention and one for progression (I am foggy on how this might work).

Reviewed by Haydeh Payami

Reviewer #1 (Remarks to the Author):

We thank reviewer 1 for their supportive comments that the approach is novel and of great interest to the wider field.

1. How many of the eQTL and pQTLs are cis versus trans SNPs? In general, for drug target MR, the number of SNPs tends to be far lower than for other forms of MR. This is not an issue, but often the focus is on cis-QTLs which are more proximal to the gene of interest, and are likely to have fewer pleiotropic effects. Furthermore, MR-Egger and other sensitivity analyses are more rarely useful because the number of SNPs and the variation in the instrument-exposure association can be very low, leading to very low power. *I don't think major changes are needed for this, but it might be good to note the number and proportion of cis versus trans SNPs you're using.*

All eQTLs used are acting in cis and located within 5 kb of the associated gene to maximise the specificity of the eQTL (page 6, lines 81-82). Since we have relatively few pQTLs, we chose to include all pQTLs available. We have included a sentence to more fully explain which pQTLs are cis vs trans SNPs (page 24, lines 392-395). We have displayed the number of SNPs per gene in our figures and the supplementary tables. The below texts are now included in the manuscript:

“We kept eQTLs with false discovery rate (FDR) < 0.05 and located within 5 kb of the associated gene to increase the specificity of the eQTL.”

“All pQTLs were found on the same chromosome as the associated gene except for: rs62143198 for PYGL, rs62143197 for QDPR, rs4253282 for GPNMB, rs2731674 for GPNMB³⁶. These latter four SNPs are therefore acting in trans.”

2. MR studies of disease progression can suffer from collider bias, in which factor which affects the incidence of disease can appear spuriously associated with progression in MR studies. How did you handle this, and can you be certain that your results for disease progression variants are not due to collider bias? Note - you should be able to address this just in the discussion - for example, by noting the difference in factors and genes identified for incidence and progression makes this explanation unlikely. It may also be worth flagging this as a potential limitation in the discussion *“for future research”*, as there are methods that are/have been developed for this issue.

Thank you for reminding us to address collider bias and adding these very helpful references (now references 49 and 50). We agree that it is important to address this, and have added the below paragraph to do so in the discussion (page 20, lines 309-318).

“In addition, progression studies are particularly affected by collider bias⁴⁹⁻⁵¹. For example, if expression of a gene and depression are both associated with disease risk, that gene's expression will be artificially associated with depression in a cohort containing only cases. In a progression study, genetic variants that cause disease will thus be associated with other risk factors for disease. The druggable genes we identified in our progression study did not reach significance in our risk study, so this kind of collider bias is less likely to have occurred for our candidate genes. The age at onset analysis is comparably more affected, since we tested genes that reached significance for PD risk. Overall, this emphasises the importance of MR quality control methods (including replication) for identifying reliable causal effects, representative

sampling in GWAS, as well as continued development of methods to formally test for collider bias^{52,53}.”

3. *It's great you've a GitHub repo for this. Given all the data are summary data, it might be worth uploading this to GitHub as well. But again, totally up to you.*

We used subsets of the public GWAS data for PD risk, and these datasets were not made openly available by the original authors. Due to these constraints of dataset access we are currently not able to upload the full input data. We have clarified our Data Availability section accordingly (page 27, lines 464-473). As reviewer 1 notes, we have made the code openly available to improve transparency of our methods and for replication in new datasets.

“The supplementary information contains the full results of this study. Any additional data may be accessed through the corresponding author upon reasonable request. Tissue-specific eQTL data were obtained from the eQTLGen (<https://eqtlgen.org/>) and PsychENCODE consortia (<http://resource.psychencode.org/>). The druggable genome data were obtained via personal correspondence with the original authors; an immediately accessible version of the druggable genome is available in the supplementary materials of the original publication⁷. Subsets of the most recent PD risk GWAS were obtained directly from the original authors⁵⁸. GWAS summary statistics for PD age at onset and an immediately accessible version for PD risk are available on the IPDGC website (<http://pdgenetics.org/resources>). Parkinson's progression GWAS data can be found here: <https://pdgenetics.shinyapps.io/pdprogmetagwasbrowser/>.”

4. How many SNPs were excluded due to Steiger filtering?

In Steiger filtering, genes are removed if the SNPs explain a greater proportion of variation in the outcome than the exposure. For the eQTL analysis, the Steiger filtering excluded 0 to 403 genes per outcome tested in a tissue, representing 0-15% of all genes tested per outcome tested in a tissue. We have added this information to the manuscript (page 26, lines 435-436).

“For the eQTL analysis, the Steiger filtering excluded 0 to 403 genes per outcome tested in a tissue, representing 0-15% of all genes studied per outcome tested in a tissue.”

Reviewer #2 (Remarks to the Author):

We thank reviewer 2 for their constructive feedback on our manuscript.

1. The authors agree (2nd sentence in discussion) that formal colocalization evidence is needed to reliably call drug targets from MR, but they have not done this in this study (apart from citing colocalization evidence for CD38 and GPNMB from earlier pQTL studies). How many targets are still left from the broader candidate set when an appropriate colocalization cut-off (e.g. $R^2 > 0.7$) is being used?

We thank reviewer 2 for this helpful suggestion to include a colocalization analysis, which we have now added (page 10, lines 161-173; page 11, lines 187-189; page 14, line 216-217; page 27, lines 451-463). Briefly, 10 of our replicated genes for PD risk and 1 gene for age at onset

had strong evidence of colocalization ($PPH4 \geq 0.75$). Our discovered genes in the progression study did not have sufficient power to perform a reliable colocalization analysis.

2. Pleiotropy is disregarded in the decision what constitutes a potential drug target. The manuscript only tests a selected set of instruments on “druggable” PD genes, but is not thoroughly investigating what effect the eQTLs and pQTLs may have on the expression of other genes nearby or in trans (be they part of the “druggable” genome or not).

If a QTL has an effect on more than one gene, this will manifest as pleiotropy in the MR estimate. As the reviewer rightly suggests, pleiotropy is a critical source of bias in MR. If the MR estimate is driven by pleiotropy, this should be apparent in the quality control (using several meta-analysis methods, MR-Egger intercept test, Cochran’s Q test and I^2 test), no matter what is causing the pleiotropy. We chose these methods because they lend themselves well to the clumping threshold $r^2 = 0.2$ (where you have to take linkage disequilibrium into account), and because they test for both measured and unmeasured pleiotropic effects. We have added sentences to explain this more thoroughly (please see our response to comment 10 by reviewer 2; page 9, 134-145). In the first paragraph of the discussion we have added the below sentences, stating which genes qualify as the most robust (page 17, lines 250-259).

“The use of several meta-analysis methods (IVW, MR-Egger and maximum likelihood), MR-Egger intercept test, Cochran’s Q test, I^2 test and colocalization allows us to look for pleiotropy due to both measured and unmeasured confounders²⁴. We consider those that pass MR quality control, have colocalization evidence or reach significance using several PD outcomes or QTL types as having the most robust MR evidence: CD38 (risk, age at onset, colocalization evidence for risk), CTSB (risk, age at onset, colocalization evidence for risk, supportive evidence using pQTLs for risk), GPNMB (risk, colocalization evidence for risk, some supportive evidence using pQTLs for risk), MAP3K12 (risk), ADAM32 (Hoehn and Yahr), IRAK3 (dyskinesia, UPDRS parts 2 and 4, depression), LMAN1 (UPDRS part 2, dyskinesias), QDPR (UPDRS part 4) and RHD (dyskinesia).”

3. The paper only concentrates on efficacy signals for PD but does not consider safety aspects, alternative indications or indirect mechanisms which also need to be considered when proposing and validating the suitability of a gene as drug target. For instance a quick look-up in the Open Targets genetics platform indicates that the GPNMB trans-pQTL rs4253282 near the F11 gene might also be associated with a range of blood clotting and vascular endpoints that may independently modulate PD risk (instead of impacting PD risk through modulating GPNMB levels). PheWAS data for these types of analyses are publicly available, e.g. from GWAS catalog, UK Biobank or FinnGen and a critical component of the target evaluation process (if the main emphasis of the paper is on nominating drug targets as it is currently being pitched).

Indeed, we focus our research on finding drug targets that have a causal relationship with a PD trait. We do agree with the reviewer that it is important to consider side effects in a drug development study, however scanning PheWAS data for potential side effects would not identify whether there is causality here. Proposing side effects with openly available PheWAS data would require an MR approach with all possible side effects as outcomes. The MR method is limited in power and unlikely to be sensitive enough to pick up side effects completely enough to make a judgement on the drug’s impact on a patient’s experience. Additionally, many medications cause significant side effects through off-target mechanisms, and such a

study would only be able to identify on-target effects. A previous MR study by Sofat and colleagues illustrates this point well (PMID: 20026784). Nevertheless we thank the reviewer for this very important point – safety considerations are of course crucial to the drug development process. We now point out that our approach does not address this (page 19, lines 296-297).

“our approach is not well suited for systematically evaluating the safety aspects of our proposed candidates in this study.”

4. MR methods accounting for LD between instruments should be employed rather than relying on LD with an arbitrary/lenient R^2 of 0.2 or distance-based cut-offs.

We thank reviewer 2 for highlighting that this was not clear, and we have attempted to explain this more fully on page 6 (lines 83-90).

“eQTLs were available for 2,786 and 2,448 druggable genes in blood and brain tissue, respectively, and these were clumped at $r^2 = 0.2$. Compared to a lower clumping threshold, this increases the number of SNPs available per gene, which in turn improves the power to detect an effect and makes it possible to test for biases in the MR estimate (as discussed later). When clumping at $r^2 = 0.2$, SNPs are not strictly independent. We therefore used MR methods that incorporate a linkage disequilibrium matrix based on the 1000 genomes EUR reference panel in the MR analysis, which accounts for correlation between SNPs^{19,20}. These methods therefore take linkage disequilibrium into account.”

5. It is not clear to me how exactly multiple testing corrections have been performed throughout the analyses. FDR is being mentioned, but has this been applied to risk or progression markers or both, and genome-wide or just for the candidate set. Please elaborate.

We agree that this was not clear in the original manuscript, and we thank the reviewer for pointing this out. We have clarified this on page 26 (lines 443-446).

“FDR-corrected p-values were calculated for each outcome to correct for multiple testing. For the discovery study for PD risk and the PD progression studies, we considered FDR < 0.05 significant. For the replication studies for PD risk and age at onset, as well as the pQTL study, we considered nominal $p < 0.05$ significant.”

6. By how much does clumping of just 2-3 instruments really improve MR estimates? Does it improve the accuracy at all and justify why this authors focus their attention on genes supported by > 1 eQTL?

It is crucial to determine whether the MR result satisfies the core assumptions of MR (now stated on page 9, lines 134-145, as displayed in our response to comment 10 by reviewer 2). This is assessed by using several methods to calculate the MR estimate (IVW, MR-Egger and maximum likelihood), as well as the MR-Egger intercept test, the Cochran’s Q test, and the I^2 test. These quality control methods are only possible when there are > 2 SNPs available, and clumping at $r^2 = 0.2$ makes it possible to complete these tests. For PD risk, our replication step also strengthens the evidence of our candidate genes.

7. Psychencode is single source of brain eQTLs in this manuscripts. While laudable that eQTLs are being tested from a tissue probably most relevant to PD, the small sample size may challenge e.g. directionality estimates which are critical for choosing the therapeutic approach. For instance, a claim that the HSD3B7 shows a different directionality in brain and blood should be supported by independent data. Also, the caveats when relying on (probably in most cases post-mortem) brain expression data for eQTL calling should be discussed.

We agree with the reviewer that the directionality of effect is crucial to get right. While we see opposing directions of effect for HSD3B7 in both our discovery and replication data, this may be driven by SNPs that do not satisfy the MR assumptions. This is why we stress that results based on one or two SNPs should be interpreted with caution: it is not possible to perform our additional quality control here (MR-Egger intercept test, the Cochran's Q test, and the I^2 test)., We are more confident in the direction of effect where there is replication and several SNPs available. We have now clarified the impact of sample size in the QTL studies on our MR results in the discussion (page 21, lines 339-342). We have also added the very good point that there are caveats to using eQTL data from post-mortem brain tissue (page 21, lines 333-336).

“Furthermore, the sample size of our blood eQTL data ($n = 31,684$) is larger than that for brain tissue eQTLs ($n = 1,387$) and the blood pQTL study ($n = 750-4,137$). A larger sample size allows greater power to detect QTLs, meaning there are more SNPs per gene, and in turn a stronger MR result.”

“A limitation of using brain tissue is that gene expression is quantified post-mortem, and measured expression levels are influenced by RNA degradation occurring after cell death as well as transcriptional changes occurring in response to death⁵⁴.”

8. Related to point 7., repurposing of Roledumab targeting RHD is supported by brain eQTL but apparently not blood eQTL data (or I have overlooked). Roledumab acts in the blood, not the brain. Why should this be a good repurposing opportunity for PD?

We thank the reviewer for this point. We have added a sentence to point this out, use more cautious language when discussing this drug, and have moved this section to the paragraph discussing medication that are less suitable for repurposing (page 18, lines 273-277).

“Other medications may not be as suitable for repurposing. To our knowledge, there is no evidence linking PD and the drug roledumab, which is currently in a phase II clinical trial to prevent alloimmunisation in Rhesus negative mothers carrying a Rhesus positive child (NCT02287896). Our evidence suggests that RHD expression in brain tissue, rather than blood, is associated with PD dyskinesia.”

9. p.3 “SNPs associated with expression levels of a gene (expression quantitative trait loci, eQTLs) are analogous to lifelong exposure to a medication targeting the encoded protein”. In this context, “are” should be substituted by “may be”, but this sentence is exemplary for the remainder of the manuscript that needs to be carefully scrutinized for overstatements.

We have added more cautionary language in our in our interpretations throughout the manuscript (e.g. addressing collider bias, exchanging: “is” with “may be”, “most robust” with “more robust”, “robustly validate” with “validate”, “minimise” with “reduce”).

10. Throughout the manuscript, the limitations of MR to predict drug targets are insufficiently highlighted. In addition to highlighting in the discussion general challenges (like lifelong genetic modulation vs acute therapeutic modulation, Europeans vs non-*Europeans*, *tissue*, ...) the authors need to detail better what assumptions the respective MR methods used are making and how easily they can be violated. For instance, challenges of utilizing weak instruments, the volatility of directionality based on messy gene expression data forming the foundation for eQTL calling, or that the number of instruments available for clumping methods is typically *very modest, ... may lead to faulty assumptions is not discussed. Overall, the broad claims in the manuscript are often not substantiated by sufficiently robust data.*

We thank the reviewer for this feedback. We have added a description of the assumptions of our MR methods and the ways these can be violated (page 9, 134-145).

“The IVW method assumes that (1) the genetic variant(s) must be associated with the exposure, (2) the genetic variant(s) must not be associated with any confounders, and (3) the genetic variant(s) must not be associated directly with the outcome. This means that the SNP should affect the outcome (PD risk) through the exposure (gene expression) only, so the y-intercept of the IVW regression is fixed at zero²⁴. This assumption is violated if there is genetic pleiotropy, where a SNP affects the outcome through an alternative pathway. This kind of pleiotropy may arise due to measured and unmeasured confounders, for example if the SNP is an eQTL for another gene that is not tested in this MR study. If pleiotropy pushes the effect in one direction, the IVW method yield a biased effect estimate. The MR-Egger method relaxes this assumption by not constraining the y-intercept. If the MR-Egger y-intercept significantly deviates from zero, this suggests that there is directional pleiotropy. This method assumes that any pleiotropic effects are independent of the gene-exposure association²⁵.”

Reviewer #3 (Remarks to the Author):

We thank reviewer 2 for their thoughtful comments and references, which we have added (references 42, 55 and 56).

1. *NDUFAF2 did not pass Cochran's Q nor I2 test. This statement is made in the context of pleiotropy. Please explain what this finding means and how it should be interpreted (I know what Q and I2 are used for, and what pleiotropy is, but don't understand how the former can inform on the latter). The usual use of Cochran's Q and I2 is to test heterogeneity. I cannot find in methods how these tests were applied here. If in the usual context, then it would imply NDUFAF2/metformin result is highly heterogenous (per statistics in supplement). Does that mean metformin may be a useful preventative drug for a subset of PD? That begs the question: which subset? This paper does not address the heterogeneity in PD. I suggest they do, and discuss how their analyses deal with the unaccounted heterogeneity in PD.*

We have expanded on what heterogeneity means in this context of this finding (page 10, lines 158-160; page 18, lines 264-267). It is a very interesting idea that heterogeneity in our MR results may occur if the effect is driven by a subset of PD patients, and we have added this point in our discussion about NDUFAF2.

“The Cochran’s Q and I^2 tests usefully assess overall heterogeneity between Wald ratios. NDUFAF2, WNT3 and VKORC1 did not pass the Cochran’s Q ($p < 0.05$) nor I^2 ($I^2 > 0.50$) tests (Supplementary Table 2). This means that there is significant heterogeneity in the MR result for these genes, and such heterogeneity among Wald ratios can for example happen if at least one SNP for the gene is pleiotropic²⁹.”

“NDUFAF2 encodes a subunit of a target of metformin, an approved medication for type 2 diabetes mellitus. There is extensive evidence for a relationship between diabetes and PD⁴⁰, and several rodent studies have investigated the potential of metformin as a neuroprotective agent⁴⁰⁻⁴². We found significant heterogeneity in the MR result for this gene. Although this may be because at least one SNP for this gene is pleiotropic, we speculate that this could occur if the effect is driven by a subset of PD patients. This however remains a subject for future research, because the GWAS data used in this study is not stratified by any kind of PD subtype.”

2. It is not surprising that there is little overlap between risk and progression, as highlighted as surprising even in abstract. They point to different things: risk points to an initial trigger (an external insult or a rare pathogenic mutation), progression points to intrinsic mechanisms that determine what happens in response to that insult. I think the more interesting question is the overlap between age at onset and progression, since the former can be viewed as the early phase of the latter. This study did not (could not with current study design) answer the overlap between age at onset and progression, but if they are amenable to additional analyses, they have the data and can attempt it.

We agree that the overlap between age at onset and progression is an interesting avenue, and we have now carried out an analysis to explore this relationship (page 14, lines 217-221). Interestingly, none of the genes that reached significance for a progression outcome reached significance in the age at onset data, and we have edited the discussion to address this (page 19, lines 284-293).

“Since age at motor symptom onset may be considered an early marker of PD progression, we also used our MR approach to assess whether genes discovered by our progression analysis causally predict age at onset. Of the genes that reached significance for a progression marker, none reached nominal significance using the age at onset data (Supplementary Table 3 and 4).”

“The two-sample MR design allows us to explore different tissues and PD traits, and we identify different candidates to prevent, delay onset, and slow progression of PD (Fig. 2, Fig. 3 and Fig. 4). Although we find that four of the drug targets for PD risk may also affect PD age at onset, we find very different candidates for progression. Age at motor symptom onset can be considered an early sign of PD progression, and it is striking that none of the genes that reached significance for a progression outcome reached significance in the age at onset data. These findings are in line with the GWAS data, finding little overlap between loci associated with PD risk, age at onset and progression markers⁹⁻¹¹. This may reflect the limited sample size of current PD progression GWAS data. Nevertheless, this raises questions about what drives PD susceptibility versus progression, painting a yet unclear picture of partially overlapping molecular mechanisms.”

3. Authors provide effect sizes: OR for risk, beta for age at onset and progression. How should

these values be interpreted in the context of druggability? Can one infer drug efficacy from these effect sizes, if so, how?

We concur with the reviewer's suggestion that effect sizes are very important in studies of drug target. However, current MR methods more reliably detect whether there is an effect of exposure on outcome, rather than the magnitude of the effect. The SNPs additionally mimic life-long and low-dose exposure, whereas a clinical trial would typically assess relatively short-term and high-dose exposure. Current MR methods also assume a linear relationship between exposure and outcome. For these reasons we cannot conclude that targets with e.g. a larger beta have a more pronounced effect, and drawing conclusions based on effect sizes can be misleading. We highlight this in our discussion (page 20, lines 319-323).

“Another key limitation of this study is that MR cannot fully recapitulate a clinical trial. MR mimics lifelong, low-dose exposure to a drug and assumes a linear relationship between exposure and outcome. This differs from a clinical trial, which typically investigates comparably high doses a drug over a much shorter timeframe. The MR result may therefore not directly correspond to the effect size in practice and does not perfectly predict the effect of a drug.”

4. As authors point out, the drugs for progression would be of more immediate interest than *drugs for prevention, because we cannot predict PD and don't know who should be subject to prevention.* (metformin is with risk/prevention). The analysis for progression is limited by short duration of follow-up (median ~3 years) and not having a second dataset for replication. I suggest authors take advantage of other publicly available datasets to improve their analysis of progression. PPMI for example is longitudinal, started with de novo unmedicated PD patients, and has followed them since 2013.

We agree with reviewer 3 that progression is of most immediate importance for PD drug development. To our knowledge, the data used here is the largest progression data openly available. We are not aware of a well-powered study that we could use for replication, which would have to measure progression in a similar way to the study we used. We note that PPMI is part of both the risk data and the progression data used here. We are confident that progression GWASs are growing, and a replication study will be possible in future research. We now discuss this on page 19, lines 297-301.

“To our knowledge, the data used here are from the largest openly available progression GWAS to date. We did not find any non-overlapping PD progression GWAS with sufficient power for a replication step in our progression analysis, which would have to measure progression in a similar way to the study used here.”

5. There are many caveats that call for “caution”. *Authors acknowledge this. They present the criteria they use to call a result more robust than others – they are subjective criteria.* I disagree with using association with risk and age at onset as corroborating evidence of association with two phenotypes – in reality they are distinct, but the way we measure them and treat them statistically, they can mimic each other (association with higher risk often also presents as earlier onset, and protection with delayed onset).

We agree with the reviewer's thoughts on this. We were surprised to find no overlap between the druggable genes for PD risk and progression in our study. As we note in our discussion,

this is in line with the original GWAS data, and we maintain that the complexities of the molecular mechanisms driving risk versus progression is important to keep in mind going forward. We have modified our language to acknowledge this (please see the quoted paragraph in response to comment 2 by reviewer 3's; page 19, lines 284-293).

6. *QQ plot (Figure S1) shows alarming inflation. It is said to have “adequate agreement with the expected null distribution”. Can you explain why it looks so inflated when inflation factor is 1.1?*

This QQ plot is similar to that found in the 2014 PD risk GWAS by Nalls and colleagues. The 2019 PD risk GWAS by the same authors did not include a QQ plot, so we have reproduced one with all of the 2019 data that's openly available; it looks very similar. We have included these plots below for your reference.

The raw lambda tends to increase with larger sample size and in meta-analyses. This is why we have computed a lambda1000, which is standardized to a scale for 1,000 cases and 1,000 controls (lambda1000 = 1.017). This was also done in the 2014 and 2019 GWAS, which found similar results. The 2019 GWAS similarly reported a raw lambda estimate of 1.170, and a lambda1000 of 1.002 (supplementary material page 5 of Nalls et al 2019; PMID: 31701892). The 2014 GWAS reported a lambda1000 that “ranged from 0.889 to 1.956” (Nalls et al 2014; PMID: 25064009). We now mention this on page 25, lines 410-413.

“Based on genomic inflation factors and quantile-quantile plots of the original GWASs^{21,58}, we considered our quantile-quantile plot to show adequate agreement with the expected null distribution (Fig. S1).”

7. Need data dictionary for supplemental tables. Supplementary material can use a thorough cleaning-up (ex. Nalls et al is not a phenotype).

We thank the reviewer for this feedback. We have made some changes to the supplementary material and added a data dictionary (Supplementary Table 8) to clarify the meaning of the terminology and abbreviations used.

Reviewers' Comments:

Reviewer #1:

Remarks to the Author:

Thank you for considering my comments, congratulations again on a really nice paper, which will make an excellent addition to the literature.

Reviewer #2:

Remarks to the Author:

1. The authors have addressed several of my concerns, which is appreciated. However, while I acknowledge that some reported findings have now been more carefully rephrased, colocalization analyses have now been conducted more broadly, and replication analyses and additional qc should reduce the risk of incorrect findings, several statements continue to make the paper seem more than it actually holds. Especially when targeting audiences in drug discovery that might consider to initiate laborious and expensive experiments based on the authors' results, the manuscript in its current form will require additional careful re-writing (see my original comments #9 and #10) as well as further analyses.

2. For instance, in the abstract the authors still state that there would be "remarkably little overlap between our drug targets to reduce PD risk versus progression", but as they admit in their response to my comment #1, none of their progression MR findings could be validated through colocalization. So, there are no progression drug targets. Likewise, nomenclature such as in Fig.1 drug targets "identified", "replicated" and "validated" should be highlighted as "potential", especially as long as there is no evidence that eQTL/pQTLs and PD GWAS sentinel SNPs colocalize. The level of confidence - or (in many cases) lack thereof - for individual findings need to be better highlighted in Figures, corresponding text and discussion.

3. Pleiotropy (comment #2) continues to be disregarded in the decision what constitutes a potential drug target. The manuscript only tests a selected set of instruments on "druggable" PD genes, but is not thoroughly investigating what effect the tested eQTLs and pQTLs may have on the expression of other genes nearby or in trans (be they part of the "druggable" genome or not).

4. For instance, the authors respond to my original comment #2 via that they would have conducted other MR approaches such as Egger tests. However, Egger only assumes net directional pleiotropy, for many will be underpowered, and comes with various additional assumptions and limitations. For instance, Egger is very limited when there is only a small number of variants as one will be estimating slope and intercept based on only a small number of data points which may render conclusions unreliable. For instance, already a quick lookup of the top WNT3 SNP in eQTLGen showed pleiotropy (see below Table). It should not be too difficult for the authors to conduct such additional analyses more broadly and with this further refine their target proposals.

SNP Gene Allele

P-value ID Chr Pos (hg19) ID Symbol Chr Pos (hg19) Z-score Assessed Other Nr Cohorts Nr Samples FDR

3.2717e-310 rs199520 17 44853872 ENSG00000238083 LRRC37A2 17 44610946 41.5701 G A 14 6421 0

3.2717e-310 rs199520 17 44853872 ENSG00000214401 KANSL1-AS1 17 44272515 58.8084 G A 13 5502 0

6.782e-286 rs199520 17 44853872 ENSG00000262539 RP11-259G18.3 17 44337444 36.1335 G A 22 16753 0

4.3816e-236 rs199520 17 44853872 ENSG00000261575 RP11-259G18.1 17 44345231 32.8091 G A 9 3831 0

1.4202e-116 rs199520 17 44853872 ENSG00000262500 RP11-259G18.2 17 44321691 22.9518 G A
20 15928 0
2.9594e-112 rs199520 17 44853872 ENSG00000073969 NSF 17 44751432 22.5152 G A 36 28917 0
1.8079e-48 rs199520 17 44853872 ENSG00000108379 WNT3 17 44875196 14.6302 G A 34 27784 0
6.3515e-37 rs199520 17 44853872 ENSG00000185829 ARL17A 17 44625578 12.6946 G A 14

Re. authors' response to my comments #6 and #10: Scrutinizing MR assumptions should be driven by a sensible biological rationale rather than simply relying on more MR sensitivity analyses: The current tools just do not serve as plug-and-play pipelines which could be applied unquestioned across all loci, and limitations of their "sensitivity analyses" are not called out. Instead of MR, the effect of a sentinel variant on other nearby genes (e.g., 1Mb or 500Kb around the variants in question with gene expression (eQTLs) or protein expression (pQTLs) measured) can be empirically tested, which the authors have not done.

5. Several of the findings in the manuscript have meanwhile been reported by Baird et al (PMID: 33417599) as potential drug targets for PD through MR-evidence and colocalization, e.g. GPNMB and HSD3B7. That study also conducted assessments across a range of additional neurological and psychiatric traits (MR-PheWAS), with several examples supported by MR and colocalization in opposite directions for more than one indication (and thus eventual safety signals, see my comment #3). Given that for PD the starting points for both studies do not seem to be substantially different (though I acknowledge several differences, e.g. Baird et al. used eQTLs from AMP-CMC instead of PsychEncode and only assessed PD-risk, while this study here focusses on pre-described "druggable" targets rather than PD GWAS more comprehensively), the overlap of MR findings between both studies seems fairly modest - again cautioning that MR results can be misleading. For researchers who may want to work downstream of the findings like the ones reported here, it would be nice if the authors highlighted (e.g. in a Table) the strength of their findings in the light of earlier PD MR-studies (e.g. Nalls et al., 2019; Kia et al., 2020; Baird et al., 2021; and now also De Klein et al., see below, Suppl Table 12) to substantiate with orthogonal evidence which of their target proposals can now indeed be considered as strong-enough to eventually warrant substantial follow-up.

6. In my comment #7 I appreciate the use of brain eQTLs, but in the manuscript brain and blood eQTLs continue to be used exchangeably, which may have led to erroneous proposals like to repurpose Roledumab targeting RHD for PD although the drug is unlikely to work in the CNS (which in response to my comment #8 is discussed as a caveat now rather than as previously a success - but it is still listed uncommented as repurposing opportunity in Table 1). I understand that mixing blood and brain is mostly done for missing power (and 31 potential drug targets just may sound more than 17 for brain tissue alone), but since first review, a substantially larger source of brain eQTLs has become available (De Klein et al. <https://www.biorxiv.org/content/10.1101/2021.03.01.433439v2>). The authors could leverage this resource (<https://www.metabrain.nl/cis-eqtl.html>) to potentially further solidify some of their findings.

Additional comments on the revised manuscript:

7. Lines 339-342

Furthermore, the sample size of our blood eQTL data ($n = 31,684$) is larger than that for brain tissue eQTLs ($n = 1,387$) and the blood pQTL study ($n = 750-4,137$). A larger sample size allows greater power to detect QTLs, meaning there are more SNPs per gene, and in turn a stronger MR result.

I don't think one can generalize more SNPs = stronger MR. This will only be the case when the effect is true and all discovered variants have the same underlying effect.

8. Lines 161-168

Additionally, an unreliable MR result may arise from a locus where the SNP-exposure and SNP outcome associations are rooted in two distinct causal SNPs in close linkage disequilibrium²⁹. When the SNP is significantly associated with both exposure and outcome, this can be probed using

colocalization analysis³⁰. There is evidence that proteins with both MR and colocalization evidence are more likely to be successful drug targets³¹; this may simply reinforce that GWAS nominated drug targets are more likely reach approval⁴. Using the discovery outcome data, we had sufficient power ($PPH3+PPH4 \geq 0.8$) to perform a colocalization analysis for 13 genes (see Methods and Supplementary Table 7).

The authors need to better clarify what they understand as “unreliable MR”. When a QTL is significantly associated with both exposure and outcome, but doesn’t show colocalization, to me that implies pleiotropy.

Reviewer #3:

Remarks to the Author:

Authors have addressed my comments, with one exception: they opted out of addressing how they envision the findings might be translated into design of clinical trials. I suppose the field will figure it out. In my opinion, while this paper does not offer any conclusive actionable results yet, it is significant because it addresses the important problem of why clinical trials for Parkinson’s fail, demonstrates a clever and novel approach to inform clinical trials, and the preliminary findings are intriguing. I hope those who design clinical trials will take note.

Haydeh Payami

REVIEWER COMMENTS AND AUTHOR RESPONSES

Reviewer #1 (Remarks to the Author):

Thank you for considering my comments, congratulations again on a really nice paper, which will make an excellent addition to the literature.

We thank reviewer 1 for their productive comments and for taking the time to review our paper.

Reviewer #2 (Remarks to the Author):

1. The authors have addressed several of my concerns, which is appreciated. However, while I acknowledge that some reported findings have now been more carefully rephrased, colocalization analyses have now been conducted more broadly, and replication analyses and additional qc should reduce the risk of incorrect findings, several statements continue to make the paper seem more than it actually holds. Especially when targeting audiences in drug discovery that might consider to initiate laborious and expensive experiments based on the authors' results, the manuscript in its current form will require additional careful re-writing (see my original comments #9 and #10) as well as further analyses.

We thank reviewer 2 for their scrutiny, and we agree that careful target validation is crucial in drug development in order to reduce the immense cost of failures. As we state in our introduction, this is a key rationale behind our approach. We have prioritised our discovered genes using well-established MR sensitivity analyses, replication, different QTL types and colocalization where possible. To make this clearer, we have produced Tables 2 and 3 to better communicate our efforts to prioritise the discovered genes, and in our abstract we have made it clearer how many genes we put forward as “high-priority”. We have also clarified this in Figure 1C.

In our discussion we have also clarified that MR approaches should operate alongside existing drug development methods. Genomic methods do not replace invaluable functional work. We hope that we have made this clearer, and we thank the reviewer for their comment.

“Of these, we put forward six drug targets with the strongest MR evidence.”

“Nevertheless, this MR study does not provide functional evidence for the proposed drug targets, and the MR process does not replace pre-clinical evaluation of drug targets *in vitro* and *in vivo*. Genomic approaches serve as an adjunct thereto, promising to better prioritise drug targets carried forward to clinical phase trials.”

2. For instance, in the abstract the authors still state that there would be “remarkably little overlap between our drug targets to reduce PD risk versus progression”, but as they admit in their response to my comment #1, none of their progression MR findings could be validated through colocalization. So, there are no progression drug targets. Likewise, nomenclature such as in Fig.1 drug targets “identified”, “replicated” and “validated” should be highlighted as “potential”, especially as long as there is no evidence that eQTL/pQTLs and PD GWAS sentinel SNPs colocalize. The level of confidence - or (in many

cases) lack thereof - for individual findings need to be better highlighted in Figures, corresponding text and discussion.

Colocalization may support a drug target if the SNP used in the MR study is significantly associated with both exposure and outcome. This is only the case if the SNP is a GWAS hit for both the exposure and the outcome. A the key assumption of MR is that the SNP is not directly associated with the outcome, and using Steiger filtering we remove any SNPs that are more strongly associated with the outcome than our exposure. Very few loci have been found in progression GWASs for Parkinson's disease. It is therefore not possible to perform a well-powered colocalization study for the genes discovered by our progression study. As such, colocalization is not an effective as a "rule-out" tool compared to replication, MR-Egger, Cochran's Q and I² tests. We hope that Tables 2 and 3 effectively display the level of confidence we have in each druggable gene.

3. Pleiotropy (comment #2) continues to be disregarded in the decision what constitutes a potential drug target. The manuscript only tests a selected set of instruments on "druggable" PD genes, but is not thoroughly investigating what effect the tested eQTLs and pQTLs may have on the expression of other genes nearby or in trans (be they part of the "druggable" genome or not).

Please see our response to comment number 4.

4. For instance, the authors respond to my original comment #2 via that they would have conducted other MR approaches such as Egger tests. However, Egger only assumes net directional pleiotropy, for many will be underpowered, and comes with various additional assumptions and limitations. For instance, Egger is very limited when there is only a small number of variants as one will be estimating slope and intercept based on only a small number of data points which may render conclusions unreliable. For instance, already a quick lookup of the top WNT3 SNP in eQTLGen showed pleiotropy (see below Table). It should not be too difficult for the authors to conduct such additional analyses more broadly and with this further refine their target proposals.

SNP Gene Allele

P-value	ID	Chr	Pos (hg19)	ID	Symbol	Chr	Pos (hg19)	Z-score	Assessed	Other	Nr Cohorts	Nr Samples	FDR
3.2717e-310	rs199520	17	44853872	ENSG00000238083	LRRC37A2	17	44610946	41.5701	G	A	14	6421	0
3.2717e-310	rs199520	17	44853872	ENSG00000214401	KANSL1-AS1	17	44272515	58.8084	G	A	13	5502	0
6.782e-286	rs199520	17	44853872	ENSG00000262539	RP11-259G18.3	17	44337444	36.1335	G	A	22	16753	0
4.3816e-236	rs199520	17	44853872	ENSG00000261575	RP11-259G18.1	17	44345231	32.8091	G	A	9	3831	0
1.4202e-116	rs199520	17	44853872	ENSG00000262500	RP11-259G18.2	17	44321691	22.9518	G	A	20	15928	0
2.9594e-112	rs199520	17	44853872	ENSG00000073969	NSF	17	44751432	22.5152	G	A	36	28917	0
1.8079e-48	rs199520	17	44853872	ENSG00000108379	WNT3	17	44875196	14.6302	G	A	34	27784	0
6.3515e-37	rs199520	17	44853872	ENSG00000185829	ARL17A	17	44625578	12.6946	G	A	14		

Re. authors' response to my comments #6 and #10: Scrutinizing MR assumptions should be driven by a sensible biological rationale rather than simply relying on more MR sensitivity analyses: The current tools just do not serve as plug-and-play pipelines which could be applied unquestioned across all loci, and limitations of their "sensitivity analyses" are not called out. Instead of MR, the effect of a sentinel variant on other nearby genes (e.g., 1Mb or 500Kb around the variants in question with gene expression (eQTLs) or protein expression (pQTLs) measured) can be empirically tested, which the authors have not done.

Testing for genetic associations with other variables is one approach to identify potential pleiotropy. Studies typically opt to test phenome-wide or transcriptome-wide assessment as a substitute for other, more established MR quality control methods when there aren't enough SNPs to perform these. Such analyses assume that all confounders have been measured. We favour the MR-Egger intercept test, Cochran's Q and the I^2 tests, as they test for pleiotropy through both measured and unmeasured variables. We thank the reviewer for providing us with this example of *WNT3*, which we had already found to be pleiotropic using the Cochran's Q and I^2 tests. We have completed a further MR analysis for our 23 proposed drug targets, removing any SNPs that are eQTLs for more than one gene in the source data.

"We repeated the analysis in the discovery outcome data with only SNPs that were specifically associated with our replicated genes. In other words, we removed any SNPs associated with the expression of any other gene in the original eQTL dataset. All replicated genes remained significant in this analysis (Supplementary Table 8, Supplementary Table 9)."

"*BST1* and *MMRN1* remained significant when removing SNPs associated with expression of any other gene in the original eQTL dataset (Supplementary Table 8, Supplementary Table 9)."

"*CD177*, *IRAK3*, *RHD* and *STK4* remained significant when removing SNPs associated with expression of any other gene in the original eQTL dataset (Supplementary Table 8, Supplementary Table 9)."

5. Several of the findings in the manuscript have meanwhile been reported by Baird et al (PMID: 33417599) as potential drug targets for PD through MR-evidence and colocalization, e.g. GPNMB and HSD3B7. That study also conducted assessments across a range of additional neurological and psychiatric traits (MR-PheWAS), with several examples supported by MR and colocalization in opposite directions for more than one indication (and thus eventual safety signals, see my comment #3). Given that for PD the starting points for both studies do not seem to be substantially different (though I acknowledge several differences, e.g. Baird et al. used eQTLs from AMP-CMC instead of PsychEncode and only assessed PD-risk, while this study here focusses on pre-described "druggable" targets rather than PD GWAS more comprehensively), the overlap of MR findings between both studies seems fairly modest - again cautioning that MR results can be misleading. For researchers who may want to work downstream of the findings like the ones reported here, it would be nice if the authors highlighted (e.g. in a Table) the strength of their findings in the light of earlier PD MR-studies (e.g. Nalls et al., 2019; Kia et al., 2020; Baird et al., 2021; and now also De Klein et al., see below, Suppl Table 12) to substantiate with orthogonal evidence which of their target proposals can now indeed be considered as strong-enough to eventually warrant substantial follow-up.

We thank reviewer 2 for their thoughts on the more recent literature, and we have added a discussion how our analysis differs from previous MR projects studying PD using QTL data. We have also included this in Tables 2 and 3.

“Our study has valuable advantages compared to previous MR studying PD using QTL data. In the latest GWAS meta-analysis for PD risk in Europeans, Nalls et al selected SNPs associated with PD risk and used MR to identify whether any of these loci act by altering expression or methylation of genes within 1 Mb of the SNP⁹. This contrasts with our exposure-centred MR analysis, where we chose SNPs associated with the expression of a druggable gene, rather than the disease outcome. More recently, Baird and colleagues conducted a transcriptome-wide MR study for a series of brain diseases and found six genes whose expression in brain tissue was significantly associated with PD risk²⁴. Two of these were also discovered in our study: *GPNMB* and *HSD3B7*. The remaining four were either not part of the druggable genome, rendering the encoded proteins less actionable drug targets, or did not reach significance in our discovery or replication cohorts, illustrating the importance of replication. Furthermore, our MR study is the first to studying druggable genes in the context of PD age at onset and progression.”

6. In my comment #7 I appreciate the use of brain eQTLs, but in the manuscript brain and blood eQTLs continue to be used exchangeably, which may have led to erroneous proposals like t repurpose Roledumab targeting RHD for PD although the drug is unlikely to work in the CNS (which in response to my comment #8 is discussed as a caveat now rather than as previously a success - but it is still listed uncommented as repurposing opportunity in Table 1). I understand that mixing blood and brain is mostly done for missing power (and 31 potential drug targets just may sound more than 17 for brain tissue alone), but since first review, a substantially larger source of brain eQTLs has become available (De Klein et al. <https://www.biorxiv.org/content/10.1101/2021.03.01.433439v2>). The authors could leverage this resource (<https://www.metabrain.nl/cis-eqtls.html>) to potentially further solidify some of their findings.

We have repeated our analysis using the De Klein et al “MetaBrain” eQTL data from cortical and basal ganglia samples and the Parkinson’s disease risk discovery data. We found that there are 1,257 and 107 druggable genes with an eQTL in the cortical data and the basal ganglia data, respectively. We have only included the PsychENCODE analysis in the manuscript for several reasons. Firstly the PsychENCODE dataset covers 2,448 druggable genes. Since our aim in drug discovery, it is crucial to cover as many druggable genes as possible.

Second, all druggable genes had between 1 and 4 eQTLs after clumping in the MetaBrain data, whereas the number of eQTLs per gene in the PsychENCODE analysis ranged from 1 to 29. As we demonstrate in the manuscript, having several SNPs per gene is crucial for MR quality control and evaluating the robustness of the result.

Third, as the PsychENCODE data is a subset of the MetaBrain resource, the MetaBrain data cannot serve as a replication step for our analysis. We note that we find some overlap between the results for Parkinson’s disease risk using the PsychENCODE versus MetaBrain data. We have attached the results from our MetaBrain analysis as an extra file for the reviewers’ and editors’ reference.

Additional comments on the revised manuscript:

7. Lines 339-342

Furthermore, the sample size of our blood eQTL data (n = 31,684) is larger than that for brain tissue eQTLs (n = 1,387) and the blood pQTL study (n = 750-4,137). A larger sample size allows greater power to detect QTLs, meaning there are more SNPs per gene, and in turn a stronger MR result.

I don't think one can generalize more SNPs = stronger MR. This will only be the case when the effect is true and all discovered variants have the same underlying effect.

We have removed the words "and in turn a stronger MR result" from this sentence.

8. Lines 161-168

Additionally, an unreliable MR result may arise from a locus where the SNP-exposure and SNP outcome associations are rooted in two distinct causal SNPs in close linkage disequilibrium²⁹. When the SNP is significantly associated with both exposure and outcome, this can be probed using colocalization analysis³⁰. There is evidence that proteins with both MR and colocalization evidence are more likely to be successful drug targets³¹; this may simply reinforce that GWAS nominated drug targets are more likely reach approval⁴. Using the discovery outcome data, we had sufficient power ($PPH3+PPH4 \geq 0.8$) to perform a colocalization analysis for 13 genes (see Methods and Supplementary Table 7).

The authors need to better clarify what they understand as "unreliable MR". When a QTL is significantly associated with both exposure and outcome, but doesn't show colocalization, to me that implies pleiotropy.

We have replaced the word "unreliable" with "spurious" here, to indicate that two distinct causal variants can lead to a false positive result.

Reviewer #3 (Remarks to the Author):

Authors have addressed my comments, with one exception: they opted out of addressing how they envision the findings might be translated into design of clinical trials. I suppose the field will figure it out. In my opinion, while this paper does not offer any conclusive actionable results yet, it is significant because it addresses the important problem of why clinical trials for Parkinson's fail, demonstrates a clever and novel approach to inform clinical trials, and the preliminary findings are intriguing. I hope those who design clinical trials will take note.

Haydeh Payami

We thank reviewer 3 for their highly constructive feedback, and we hope that the prioritization tables we have added make our results more actionable.

Reviewers' Comments:

Reviewer #2:

Remarks to the Author:

I thank the authors for taking into account my remaining concerns. I would like to particularly applaud them for the introduction of Tables 2 and 3 which indeed provide helpful guidance as to what is the level of confidence vs remaining uncertainty for each proposed gene individually. To highlight the absence of GWAS evidence, and with this weaker overall confidence into respective results, I would encourage them to leave an (empty) "Colocalization" column also for PD progression in Table 3 (my point 2).

Heiko Runz

REVIEWERS' COMMENTS

Reviewer #2 (Remarks to the Author):

I thank the authors for taking into account my remaining concerns. I would like to particularly applaud them for the introduction of Tables 2 and 3 which indeed provide helpful guidance as to what is the the level of confidence vs remaining uncertainty for each proposed gene individually. To highlight the absence of GWAS evidence, and with this weaker overall confidence into respective results, I would encourage them to leave an (empty) "Colocalization" column also for PD progression in Table 3 (my point 2).

Heiko Runz

We thank the reviewer for their supportive comments, and we have included a "Colocalization" column in Table 3.